



# Formaldehyde and hydroperoxide distribution around the Arabian Peninsula – evaluation of EMAC model results with ship-based measurements

Dirk Dienhart[1], Bettina Brendel[1], John N. Crowley[1], Philipp G. Eger[1], Hartwig Harder[1], Monica Martinez[1], Andrea Pozzer[1], Roland Rohloff[1], Jan Schuladen[1], Sebastian Tauer[1], Jos Lelieveld[1,2], and Horst Fischer[1]

[1]Atmospheric Chemistry Department, Max Planck Institute for Chemistry, Mainz, Germany
[2]Energy, Environment and Water Research Center, The Cyprus Institute, Nicosia, Cyprus

*Correspondence to*: Dirk Dienhart (D.Dienhart@mpic.de) or Horst Fischer (Horst.Fischer@mpic.de)

**Abstract.** Formaldehyde (HCHO) and hydrogen peroxide ($H_2O_2$) play a key role in atmospheric oxidation processes. They act as sources and sinks for $HO_x$ radicals ($OH + HO_2$), with OH as the primary oxidant that governs the atmospheric self-cleaning capacity. Measurements of these species allow evaluation of global chemistry-transport models which need to account for multifarious source distributions, transport and mixing, complex photochemical reaction pathways and deposition processes. HCHO is an intermediate produced during the oxidation of VOCs and is an indicator of photochemical activity and combustion related emissions. Due to its many production pathways and its rather short lifetime of only several hours at noon, accurate modelling of this species is challenging. In this study, we use in situ observations in the marine boundary layer (MBL) to evaluate results of the general circulation model EMAC (ECHAM5/MESSy2 Atmospheric Chemistry). The dataset was obtained during the AQABA ship campaign around the Arabian Peninsula in summer 2017. This region is characterized by high mixing ratios of photochemical air pollution, high humidity and strong solar irradiation, especially in the area around the Suez Canal and the Arabian Gulf. We find that EMAC fails to predict absolute mixing ratios of HCHO, especially during high pollution events, but it reproduces most of the HCHO variability seen in the different regions, while it systematically overestimates $H_2O_2$. This is mainly attributed to missing primary VOC emissions and the overestimation of $HO_x$ radicals, and also related to the models coarse spatial resolution.

## 1 Introduction

The effects of anthropogenic emission of greenhouse gases and aerosols and their increasing impact on climate and air quality represent a global threat. Industrialization enabled the economic development to the modern society, which is characterized by urbanization and immense population growth. Large shares of the agriculture and industry are coupled to the utilization of fossil fuels and thus, emission controls and the characterization of air quality and its health impacts are of increasing importance. Globally, fossil-fuel-related emissions account for about 65 % of the excess mortality and 70 % of the climate





cooling by anthropogenic aerosols (Lelieveld et al., 2019). Worldwide some of the largest oil reservoirs are being mobilized by the oil and gas industry in the Middle East, increasing also the local ship traffic drastically. The emission of volatile organic compounds (VOCs) and nitrogen oxides ($NO_x = NO + NO_2$) by combustion processes make this region a hotspot of tropospheric air pollution in the last decade and favor the production of tropospheric ozone ($O_3$). The Arabian Peninsula is overall characterized by unique atmospheric conditions (e.g. high temperatures and intense solar irradiation, accompanied by

aridity, low cloudiness and occasional dust storms), which classifies the region as a unique environment to study the abundance of atmospheric pollutants and their processing through photochemical oxidation.

The oxidation capacity of the atmosphere determines its self-cleaning ability and is mainly controlled by hydroxyl (OH) radicals in the gas phase. OH oxidizes methane ($CH_4$) and other VOCs so that these gases are efficiently removed from the atmosphere, e.g. by transition into the particle phase and subsequent rain-out. Lelieveld et al. (2016) showed that global OH

concentrations are buffered with a mean recycling probability of 67 %, indicating that OH is not very sensitive to perturbations by natural or anthropogenic emission changes. This buffering mechanism is based on complementary primary and secondary production of OH, e.g. through photo dissociation of ozone ($O_3$), reservoir species and radical recycling mechanisms (Lelieveld et al., 2016). OH recycling is generally dominated by the reaction of peroxy radicals with NO, which is referred to as the $NO_x$ recycling mechanism of OH.

Besides hydroxyl radicals, peroxides are a main contributor to the oxidation capacity of the atmosphere, especially in the liquid phase. Further, $H_2O_2$ plays a key role in atmospheric sulfate formation and acts as a temporary reservoir for OH. With its lifetime of several hours, $H_2O_2$ enables horizontal or vertical transport of $HO_x$ by e.g. advection/convection of air masses (Nussbaumer et al., 2021a). However, $H_2O_2$ also transitions readily into the liquid phase and thus also acts as a net sink for $HO_x$ radicals via its dry and wet deposition. To understand the $H_2O_2$ budget and its diurnal variability, it is necessary to consider

all physical and chemical processes within the atmosphere. Besides the net photochemical production (production minus loss) and deposition, horizontal and vertical transport have to be considered. The variation of the $H_2O_2$ mixing ratio in the absence of clouds during the day can be described by Eq. 1 (Fischer et al., 2019):

$$\frac{d[H_2O_2]}{dt} = P_{chem} - L_{chem} + \frac{\omega_e \Delta[H_2O_2] - V_{Dep}[H_2O_2]}{h_{BL}} - \nabla(v[H_2O_2]) \qquad (1)$$

with $P_{chem}$ as the sum of all photochemical production terms and $L_{chem}$ the sum of photochemical losses. The third term

describes vertical transport in the well mixed boundary layer, which is determined by entrainment and deposition. $\omega_e$ represents the entrainment velocity with $\Delta[H_2O_2]$ as the concentration difference between the boundary layer and the free troposphere. The deposition is determined by the deposition velocity ($V_{Dep}$) and the boundary layer height ($h_{BL}$). The last term describes the effect of horizontal transport on the $H_2O_2$ budget due to a gradient in $H_2O_2$ mixing ratios ($-\nabla(v[H_2O_2])$).

The dominant photochemical source of $H_2O_2$ is the recombination of $HO_2$ radicals which involves a collisionpartner (M)

usually nitrogen ($N_2$), oxygen ($O_2$) or water vapor ($H_2O$):

$HO_2 + HO_2 + M \rightarrow H_2O_2 + O_2 + M.$ \qquad (R1)



The production of $H_2O_2$ via R1 competes with the reaction of nitrogen monoxide (NO) and $HO_2$ (R2), which is one of the most important reactions in the troposphere to recycle OH radicals (Lelieveld et al., 2016).

$$HO_2 + NO \rightarrow OH + NO_2 \tag{R2}$$

The photochemical formation of peroxides therefore depends to a large extent on the abundance of $NO_x$, as elevated mixing ratios of NO (~ 100 pptv and higher) substantially suppress peroxide formation (Lee et al., 2000). Photochemical loss reactions of $H_2O_2$ are the conversion by OH radicals to $HO_2$ radicals (R3) and photolysis as a source of OH (R4).

$$H_2O_2 + OH \rightarrow HO_2 + H_2O \tag{R3}$$

$$H_2O_2 + h\nu \rightarrow 2\,OH \tag{R4}$$

Note that R3 and R4 regenerate $HO_x$, and thus only physical removal of $H_2O_2$ from the atmosphere establishes a net loss of $HO_x$. According to its relatively high Henry's law coefficient (~ $10^5$ mol L$^{-1}$ atm$^{-1}$), $H_2O_2$ is highly soluble and thus efficiently washed out by rain or fog (Klippel et al., 2011; Fischer et al., 2019). Dry deposition also contributes significantly to the removal of $H_2O_2$ in the boundary layer with typical deposition velocities of 0.1 – 5 cm s$^{-1}$ (Stickler et al., 2007; Nguyen et al., 2015), which leads to a local maximum of $H_2O_2$ mixing ratios above the boundary layer (Stickler et al., 2007; Klippel et al., 2011).

In the MBL, $H_2O_2$ concentration gradients are small, so that horizontal transport becomes unimportant. Additionally, the MBL height is relatively constant with no significant diel variation and thus vertical transport is weak, except close to convective clouds (Nussbaumer et al. 2021a, Fischer et al., 2015). Therefore, the $H_2O_2$ distribution in the MBL depends largely on net photochemical tendencies and deposition processes (Fischer et al., 2015). The $H_2O_2$ budget in the continental boundary layer is more complex, since all terms in Eq. 1 contribute significantly to the $H_2O_2$ budget and the boundary layer height follows a

relatively strong diel variation. In situ observations in various locations enable (together with meteorological and boundary layer height information) assesment of the role of $H_2O_2$ for the oxidizing capacity of the atmosphere (Fischer et al., 2019). Various measurement techniques have been developed to determine its vertical and geographical distribution, understand its budget and its response to natural and anthropogenic perturbations (Hottmann et al., 2020; Fischer et al., 2019; Bozem et al., 2017; Fischer et al., 2015; Klippel et al., 2011; Snow et al., 2007; Lee et al., 2000; Sauer et al., 1997).

Similar to $H_2O_2$, organic peroxides (ROOH) impact the oxidative potential of the atmosphere significantly and they also act as $HO_x$ reservoirs (Lee et al., 2000). Methyl hydroperoxide (MHP, $CH_3OOH$) is generally the most abundant gaseous, organic hydroperoxide, which is produced by the reaction of $HO_2$ with methylperoxy radicals ($CH_3O_2$) formed e.g. during the photochemical oxidation of methane ($CH_4$) (R5, R6), or by reactions of acetyl peroxy radicals ($CH_3C(O)O_2$) with $HO_2$ and NO, which can dominate the production of $CH_3O_2$ (Crowley et al. 2018). Note that the production of MHP competes with the

production of formaldehyde (HCHO, R7) from the methylperoxy radical (Nussbaumer et al., 2021b). Besides the photochemical pathways, $H_2O_2$ and MHP have also been observed to be directly released from biomass burning (Lee et al., 1997).

$$OH + CH_4 + O_2 \rightarrow CH_3O_2 + H_2O \tag{R5}$$

$$HO_2 + CH_3O_2 \xrightarrow{\sim 0.7} CH_3OOH + O_2 \tag{R6}$$



$HO_2 + CH_3O_2 \xrightarrow{\sim 0.3} HCHO + H_2O + O_2$ (R7)

The main loss reactions of MHP are its photolysis ($\sim 5 \cdot 10^{-6}$ s$^{-1}$ at sun peak) (R8), the reaction with OH (R9) (with a lifetime of ~15 hours for most regions during AQABA) and physical deposition processes, although it is ~ two orders of magnitude less soluble than $H_2O_2$ (O'Sullivan et al., 1996; Klippel et al., 2011).

$CH_3OOH + h\nu \rightarrow CH_3O + OH$ (R8)

$CH_3OOH + OH \rightarrow CH_3O_2 + H_2O$ (R9a)

$CH_3OOH + OH \rightarrow CH_2OOH + H_2O \rightarrow HCHO + OH$ (R9b)

$H_2O_2$ and MHP can be found in comparable concentrations in many parts of the atmosphere, with the highest variations in the boundary layer (Reeves and Penkett, 2003; Klippel et al., 2011). Besides MHP, peracetic acid (PAA) is another abundant organic hydroperoxide in the troposphere. PAA production rates depend on $HO_2$ and the acetyl peroxy radical ($CH_3C(O)O_2$),

which is considered one of the four most abundant organic peroxy radicals (Tyndall et al., 2001; Crowley et al., 2018). Acetyl peroxy radicals also react rapidly with $NO_x$, thus the highest concentrations of PAA are expected in regions which are impacted by biogenic emissions in which $HO_2$ levels are high enough to compete with $NO_x$ (Berasategui et al., 2020; Phillips et al., 2013). Further organic peroxides are formed in the oxidation of isoprene and other volatile organic compounds (VOCs) (Wennberg et al., 2018; St. Clair et al., 2016; Reeves and Penkett, 2003; Sauer et al., 1999; O'Sullivan et al., 1996). Recent

studies also indicate the oxidative potential of isoprene hydroxyl hydroperoxides (ISOPOOH) for sulfate formation in cloud droplets, which could even surpass that of $H_2O_2$ in forested regions (Dovrou et al., 2021; Dovrou et al., 2019).

Another major $HO_x$ reservoir is formaldehyde, which is a ubiquitous trace gas and the most abundant aldehyde in the troposphere. HCHO is highly reactive and acts as a major source of $HO_2$ via its photolysis to H and HCO radicals. It can be emitted directly from a variety of both biogenic and anthropogenic sources and is an intermediate during the oxidation of a

large number of VOCs, making budget assessments highly complex. Previous studies designed to dustinguish between secondary production and direct emissions of HCHO vary widely in their estimates and highlight the importance of local phenomena (Dienhart et al., 2021; Nussbaumer et al., 2021b, Luecken et al., 2018; Anderson et al., 2017; Wolfe et al., 2016; Stickler et al., 2006; Lee et al., 1997).

Remote sensing techniques on satellites platforms enable global observations of HCHO and thus identification of VOC

oxidation hotspots (e.g. due to oxidation of isoprene and anthropogenic emissions) and seasonal variability (Zhu et al., 2020; De Smedt et al., 2018; De Smedt et al., 2015; De Smedt et al., 2012; Marbach et al., 2009). HCHO measurements are currently used as HCHO / $NO_x$ ratios for $O_3$ sensitivity studies (i.e., $NO_x$ or VOC limitation) and global mapping of OH variability in remote air (Nussbaumer et al., 2021a; Nussbaumer et al., 2022; Tadic et al., 2020; Wolfe et al., 2019; Schroeder et al., 2017; Wolfe et al., 2016). In very clean conditions like the remote MBL or the free troposphere, HCHO production is dominated by

the photo-oxidation of methane (R5), with the bimolecular self-reaction of methyl peroxy radicals the rate limiting factor (R11) (Nussbaumer et al., 2021b; Wagner et al., 2001). The methoxy radical product ($CH_3O$) reacts quasi-instantaneously with oxygen to form HCHO and $HO_2$ (R12). In continentally influenced air masses (NO $\geq$ 100 pptv) R11 is suppressed, as methyl





peroxy radicals rapidly oxidize NO, which accelerates HCHO and simultaneously limits MHP formation (Nussbaumer et al., 2021b; Klippel et al., 2011; Lee et al., 2000).

$CH_3O_2 + CH_3O_2 \rightarrow 2\ CH_3O + O_2$ (R11)

$CH_3O + O_2 \rightarrow HCHO + HO_2$ (R12)

$CH_3O_2 + NO \rightarrow CH_3O + NO_2$ (R13)

Photolysis of the $NO_2$ product (R2, R13) leads to tropospheric $O_3$ formation. Further sources of HCHO are the photochemical degradation of several VOCs, e.g. the ozonolysis of isoprene and other alkenes as well as the degradation of MHP,

acetaldehyde, acetone and methanol (Nussbaumer et al., 2021b; Wennberg et al., 2018; Wolfe et al., 2016; Snow et al., 2007; Stickler et al., 2006).

$CH_3OH + OH \rightarrow CH_3O + H_2O$. (R14)

Since the sources of HCHO are diverse, an alternative approach instead of calculating the HCHO budget is to derive the production rate of HCHO from measurements of OH-reactivity towards VOCs, as demonstrated for the Air Quality and climate

change in the Arabian Basin (AQABA) campaign data by Dienhart et al. (2021).

Tropospheric HCHO, which is not removed heterogeneously via deposition, reacts with OH or undergoes photolysis to release $HO_2$ radicals (R15 – R17) (e.g. Heikes et al., 2001).

$HCHO + OH \rightarrow HCO + H_2O$ (R15)

$HCHO + h\nu \rightarrow H_2 + CO$ (R16a)

$HCHO + h\nu \rightarrow HCO + H$ (R16b)

$HCO + O_2 \rightarrow CO + HO_2$ (R17)

Anthropogenic release of HCHO by the oil and gas industry, biomass burning, and secondary production can significantly enhance local $HO_2$ production (Parrish et al., 2012; Klippel et al., 2011; Lee et al., 1997). Since the atmospheric lifetime of HCHO is at least several hours and it is released during the photochemical oxidation of numerous VOCs, it is a suitable tracer

for anthropogenic activity and combustion processes including biomass burning. The budget of HCHO can be described similarly to $H_2O_2$ via Eq. 1: its photochemical production pathways depend strongly on the abundance and the composition of VOCs. In the free troposphere, the main sources of HCHO are the photochemical degradation of methane, methanol and MHP (Stickler et al., 2006), whereas in the boundary layer the oxidation of alkenes (e.g. isoprene, ethene), alkanes, and the photochemical degradation of e.g. acetaldehyde, acetone, peroxyacetyl nitrate (PAN) and dimethyl sulfide (DMS) become

more significant (Crowley et al., 2018; Nussbaumer et al., 2021b). Wolfe et al. (2016) showed that the link between HCHO and isoprene oxidation is a strong, nonlinear function of $NO_x$. Primary emissions of HCHO are dominated by combustion processes, with the combustion of fossil fuels in industrialized areas (Williams et al., 2009; Wert et al., 2003) and biomass burning as a strong local source (Kluge et al., 2020, Coggon et al., 2019). Heterogeneous losses via wet and dry deposition also significantly influence the HCHO distribution, although it is less soluble than $H_2O_2$. This is also reflected in the vertical

profile, as maximum mixing ratios of HCHO are typically found in the boundary layer and decrease with altitude in the free troposphere (Zhu et al., 2020; Anderson et al., 2017; Stickler et al., 2007). In clean MBL conditions, HCHO mixing ratios





mainly depend on the abundance of $HO_x$ and it is therefore rather homogenously distributed, whereby horizontal transport is not significant. In more polluted conditions, horizontal transport can significantly influence HCHO mixing ratios on a regional scale. Vertical transport of HCHO is often limited to the MBL, as the boundary layer height is almost constant, except close

to convective clouds where elevated mixing ratios of HCHO can be used as an indicator for recent convection (Anderson et al., 2017).

In this study we present the first ship-based measurements in the marine boundary layer of the Arabian Gulf and around the Arabian Peninsula. $H_2O_2$, organic peroxides and HCHO mixing ratios were evaluated during AQABA in summer 2017 and compared to the 3-D general circulation model EMAC (ECHAM5/MESSy2 Atmospheric Chemistry). Dry deposition rates of

$H_2O_2$ and HCHO were determined during night using the method of Shepson et al. (1992). Photochemical equilibrium concentrations of $H_2O_2$ were evaluated with measured OH, $HO_2$ and actinic flux measurements.

## 2 Experimental

### 2.1 AQABA campaign

The **A**ir **Q**uality and Climate Change around the **A**rabian **Ba**sin (AQABA) measurement expedition took place from June 25

until September 3, 2017. Instrumentation of the ship (*Kommandor Iona*) was performed in La Seyne-sur-Mer (near Toulon, France), from where the first leg of the cruise started through the Mediterranean, the Suez Canal and the Red Sea to the first stop in Jeddah. The expedition continued two days later via the Gulf of Aden, the Indian Ocean, the Gulf of Oman and the Arabian Gulf (also Persian Gulf) to Kuwait. On the second leg, the ship returned with the same route (Fig. 1), without stopping in Jeddah, to end the expedition at Stromboli volcano. The *Kommandor Iona* was equipped with a weather station and five

laboratory containers on the front deck with instrumentation for in-situ and offline monitoring of a large variety of gaseous species, particles and radicals. Details about the measurements performed during AQABA can be found in a number of previous publications (Dienhart et al., 2021; Friedrich et al., 2021; Paris et al., 2021; Celik et al., 2020; Tadic et al., 2020; Wang et al., 2020; Bourtsoukidis et al., 2019; Pfannerstil et al., 2019; Eger et al., 2019;).

### 2.2 Instrumentation and sampling

HCHO and hydroperoxides were measured using modified commercial Aero-Laser instruments (AL2021, AL4021, Aero-Laser, Germany), which were placed in a temperature-controlled container. With the exception of the aerosol and radical measurements (OH and $HO_2$), air was sampled from a high-flow ($10\ m^3\ min^{-1}$) cylindrical stainless steel inlet (HFI, sampling height: 5.5 m above deck, diameter: 0.2 m), placed between the containers on the front deck of the ship. Air was drawn from the center of the HFI into the air-conditioned laboratory containers using PFA (perfluoroalkoxy alkane) tubing. The 4.2 m long

½" PFA-bypass was insulated to prevent condensation and was used with a flow rate of 12 L $min^{-1}$, which resulted in a residence time of ~ 9 s for both instruments. This setup ensured no vessel contamination while sampling against the wind



direction and minimized sampling artifacts e.g. by preventing condensation. The sampling bypass was exchanged in Kuwait before the second leg.

## 2.3 HCHO measurements

HCHO measurements were performed based on the fluorometric Hantzsch reagent method (AL4021 therefore called 'Hantzsch monitor') following the principle of (Dasgupta et al., 1988) and the design of (Kelly and Fortune, 1994). In a first step, HCHO is stripped from an airflow of 1 L min$^{-1}$ into 0.025 M H$_2$SO$_4$ (sulphuric acid for analysis, 96%, Acros Organics) with a flow of 0.55 ml min$^{-1}$ at 10°C in a stripping coil. The acidity of the stripping solution promotes quantitative solubility of HCHO and minimizes the dissolution of gaseous SO$_2$ which otherwise could interfere by formation of a sulfur adduct in the liquid phase.

Subsequently, HCHO(aq) quantitatively reacts with pentane-2,4-dione (acetylacetone, EMSURE for analysis, 99%, Merck) and ammonia (ammoniumacetate, 99%, VWR) at low pH (acetic acid, analytical grade, 100%, Serva) in the reactor coil, thermostatted at 65°C, to form the Hantzsch product 3,5-diacetyl-1,4-dihydrolutidine (DDL). DDL is subsequently detected by excitation at 410 nm with a mercury Pen-Ray® lamp, followed by collection of the fluorescence radiation 90° off axis around 510 nm with a photomultiplier tube (Hamamatsu Photonics, model H957-01). Aqueous HCHO standards were used to

calibrate the response. Line losses and sampling efficiency during the campaign were corrected by measuring gaseous standards generated using a HCHO permeation source (2.5).

## 2.4 H$_2$O$_2$ and organic hydroperoxide measurements

H$_2$O$_2$ and organic hydroperoxides (ROOH) were measured with the AL2021 based on the dual enzyme technique described in Lazrus et al. (1985). Ambient air is collected through a bypass with 2.3 L min$^{-1}$ and consequently passed through a glass coil

together with a buffered (potassium hydrogen phthalate for analysis, PanReac; NaOH, 1 mol/L, Fluka) stripping solution (0.55 mL min$^{-1}$, pH 5.8). Hydroperoxides dissolve in the stripping solution with a stripping efficiency depending on their Henry's law constant (O'Sullivan et al., 1996). Typically, H$_2$O$_2$ is dissolved quantitatively, CH$_3$OOH (methyl hydroperoxide, MHP) the smallest organic hydroperoxide, with a stripping efficiency of ~ 60% (Hottmann et al., 2020; Klippel et al., 2011). As the instrument does not differentiate between different organic hydroperoxides and as solvation is a critical step for

quantification, the AL2021 delivers a lower estimate of the total organic hydroperoxide mixing ratios. The dissolved hydroperoxides are separated into two channels and subsequently detected via reaction to a fluorescent dye with horseradish peroxidase (HRP, Sigma Aldrich) and 4-hydroxyphenylacetic acid (POPHA, Sigma Aldrich). The dimer of POPHA, 6,6'-dihydroxy-3,3'-biphenyldiacetic acid, is formed stoichiometrically and detected by fluorescence spectroscopy via excitation with a Cadmium Pen-Ray® lamp at 326 nm. Detection of the fluorescence radiation 90° off axis is performed between 400 –

420 nm with a photomultiplier tube (Hamamatsu Photonics, model H957-01) for both channels. The enzyme catalase (Sigma Aldrich) is injected into the reaction coil of channel B, prior to the reaction with HRP and POPHA, to selectively destroy H$_2$O$_2$. This technique allows quantification of H$_2$O$_2$ by calculation of the difference between channel A, which delivers the total mixing ratio of ROOH and H$_2$O$_2$, and channel B, which delivers the total mixing ratio of ROOH. Since this principle is





dependent on the catalase efficiency, it is determined for every liquid calibration and was in the range of 95 – 100% during
AQABA. In addition to the AL2021, we also operated an instrument for the detection of different organic peroxides separated
by HPLC (high performance liquid chromatography). Similar to the AL2021, it utilizes the selective dual enzyme technique
by post column derivatization and thus the HPLC enables quantification of separated organic hydroperoxides and $H_2O_2$ in low
pptv levels. When the sea was rough, the movement of the ship interfered with the instrument, causing drifts of the baseline,
which may have been caused by pressure variations within the constant-flow eluent pumps. Therefore, quantification of the
organic hydroperoxides was not possible and we only used the chromatograms for qualitative identification of the more
abundant species.

## 2.5 Calibration and instrument characteristics during AQABA

External calibration of both instruments was performed with aqueous standards (HCHO, $H_2O_2$) by dilution of stock solutions.
The $H_2O_2$ stock solution was prepared with 1 mL $H_2O_2$ (30%, Roth) in 999 mL $H_2O$ (EMSURE®, Merck) and checked for
stability by regular titration with potassium permanganate ($KMnO_4$, 0.002 mol/L, Merck). The HCHO stock solution consisted
of 3 mL HCHO (37%, Sigma Aldrich) in 997 mL $H_2O$ and was titrated against iodine ($I_2$, 0.05 mol/L, Merck).
In addition, gaseous standards were measured to calculate the inlet efficiency of the PFA-bypass. Calibration gas flows were
generated using permeation devices in temperature controlled glass flasks, which were flushed at a constant flow rate of 50
standard cubic centimeters (sccm) per minute with zero air (Zero Air generator CAP 60, Infiltec, Germany). HCHO calibration
gas was created from a paraformaldehyde container (VICI AG, Switzerland) which was heated to 60 °C. The gaseous $H_2O_2$
standard was generated from a permeation source built with a 15 cm long 1/8" polyethylene (PE) tube, that was filled with the
30% $H_2O_2$ solution, closed with PFA fittings (Swagelok, USA) and heated to 35 °C. The highly concentrated flow was then
diluted with additional zero air. The permeation rates of both sources were measured based on the chromotropic acid reaction
(Altshuller et al., 1961) and the reaction of $H_2O_2$ with $TiCl_4$ described in Pilz and Johann (1974). Note that the AL2021 has
known $O_3$ and NO interferences, which were accounted for in the final dataset. We found an interference of 36 pptv $H_2O_2$
equivalents per 100 ppbv $O_3$ and 12 pptv $H_2O_2$ per 100 ppbv NO. We did not find a significant $O_3$ interference in lab
experiments for the AL4021.
Zero gas measurements were performed every 3.5 h for 30 minutes to account for baseline drifts and to determine the
instrument's stability. For this purpose, we used a bypass via a three-way valve with a silica gel cartridge ($SiO_2$ with orange
indicator, Roth) to dry the sampled air followed by a scrubber cartridge containing hopcalite ($MnO_2/CuO$, IAC-330, Infiltec,
Germany) and platinum ($Pt/Al_2O_3$, IAC-114, Infiltec, Germany) as catalysts to destroy the remaining hydroperoxides, HCHO,
other OVOCs and $O_3$.
Both instruments log data on a custom built computing unit (V25) with a 3 second time resolution, but the data shown in this
paper was at least averaged to the so-called effective time resolution, which was determined as the response time of the
instrument (10 to 90% of the signal intensity during the injection of liquid standards). The limit of detection (LOD) was
calculated as the 2σ deviation of all zero air measurements during AQABA at the effective time resolution of 180 seconds.



The precision (P) was calculated by the $1\sigma$ deviation of the liquid standard calibrations throughout the whole measurement campaign, therefore it contains also the pipetting error during the preparation of the standards. The total measurement uncertainty (TMU) was calculated according to Gaussian error propagation. In this equation, S is the uncertainty of the
standard, IE the inlet efficiency and OI the $O_3$ interference.

$$TMU = \sqrt{(P)^2 + (S)^2 + (IE)^2 + (OI)^2} \qquad (2)$$

**Table 1: Instrument characteristics of the HCHO and hydroperoxide measurements during the AQABA campaign.**

|  | **HCHO (AL4021)** | **$H_2O_2$ (AL2021)** | **ROOH (AL2021)** |
|---|---|---|---|
| Time Res. | 180 s | 180 s | 180 s |
| LOD ($2\sigma$) | 80 - 128 pptv | 13 pptv | 8 pptv |
| P ($1\sigma$) | 1.5% @ 8.1 ppbv | 1.2% @ 4.4 ppbv | 1.7% @ 4.5 ppbv |
| TMU | 13 % | 20 % | $\geq$ 40 % |

Note that for the LOD of the AL4021, we found a significant change of the background noise, while operating the instrument when the sea was rough with strong waves. Excluding times of rough sea yields a LOD ($2\sigma$) of 80 pptv.

**2.6 Further measurements**

OH and $HO_2$ were performed with the **H**ydr**O**xyl **R**adical Measurement **U**nit based on Fluorescence **S**pectroscopy (HORUS) and sampled from a separated inlet closer to the LIF (laser-induced fluorescence) instrument, to achieve as low as possible residence times in the sampling. The instrument utilizes LIF of the OH radical, and by the titration of $HO_2$ with NO, simultaneous measurements of $HO_2$ are implemented (Hens et al., 2014; Marno et al., 2019).
Wavelength resolved actinic flux was measured with a spectral photometer (CCD Spectroradiometer 85237) to calculate photolysis frequencies (*j*-values). The radiometer was installed about 5 m above the front deck level and it was cleaned every morning to remove sea salt and dust particles. Decreases in sensitivity due to sensor contamination were corrected with a linear interpolation between the cleaning events. The *j*-values are not corrected for upwelling actinic flux from the sea surface and therefore the TMU was estimated with >10%, depending on the reaction. More details about the setup and calibration of CCD
spectroradiometers can be found in Bohn and Lohse (2017). Temperature, pressure, wind direction and speed were measured with the **Eu**ropean **C**ommon **A**utomatic **W**eather **S**tation (EUCAWS), a weather station specifically designed for ships.

**2.7 Global Atmospheric Chemistry model EMAC**

ECHAM5/MESSy2 Atmospheric Chemistry (EMAC) model is a numerical chemistry general circulation model (CGCM), which describes tropospheric and middle atmosphere processes. EMAC is based on 5th generation of the **E**uropean **C**enter
**HAM**burg (ECHAM5), a general circulation model (Roeckner et al., 2006), and uses the second version of the Modular Earth Submodel System (MESSy2) to link multi-institutional sub models (Joeckel et al., 2010). Here we use EMAC with the chemistry mechanism MOM (**M**ainz **O**rganics **M**echanism) implemented with the sub-model MECCA, which contains the



basic $HO_x$, $NO_x$ and $CH_4$ chemistry, but also halogens, sulfur and mercury (Sander et al., 2019; Lelieveld et al., 2016). Development of MOM also included a variety of NMHCs, aromatics and OVOCs including isoprene und terpene oxidation

(Sander et al., 2019) and, recently, the model has been thoroughly evaluated with this chemical mechanism (Pozzer et al., 2022). Therefore, it is ideal to test the model with the complex photo oxidation during AQABA, especially in the Arabian Gulf, where a lot of oil and gas industry is operating and model results already identified it as a hotspot of tropospheric $O_3$ (Lelieveld et al., 2009). The model simulations were carried out in the T106L31 resolution, which correspond to a grid of 1.1° x 1.1° with 31 vertical pressure layers. Previous results of airborne and shipborne expeditions have been compared to EMAC

(Fischer et al., 2015; Klippel et al., 2011), also the AQABA datasets of $NO_x$, $O_3$ and VOCs during AQABA have been published (Tadic et al., 2020; Wang et al., 2020). The dynamics have been weakly nudged (Jeuken et al., 1996; Jöckel et al., 2006) towards the ERA-interim data (Berrisford et al., 2011) of the European Centre for Medium-Range Weather Forecasts (ECMWF) to reproduce the actual day-to-day meteorology in the troposphere.

**2.8 Methods**

In the MBL, the production of peroxides is dominated by the recombination of peroxyl radicals, thus the in situ measurements of $HO_2$ enable calculation of the production rates $P(H_2O_2)$ with Eq. 4.

$$P(H_2O_2) = k_{HO_2+HO_2}[HO_2]^2 \tag{3}$$

Photochemical loss reactions are the photolysis and the reaction with OH, besides the deposition, which is the dominant loss during night.

$$L(H_2O_2) = \left( j_{H_2O_2} + k_{H_2O_2+OH}[OH] \right) [H_2O_2] + k_{Dep(H_2O_2)}[H_2O_2] \tag{4}$$

Since the measurements were performed in the MBL, the water dependency of the $HO_2$ recombination becomes significant and was calculated with the relative humidity measurements (RH) via Eq. 10 - 12 (https://iupac-aeris.ipsl.fr/test/front-office/datasheets/pdf/HOx14.pdf, July 2022).

$$k_{HO_2+HO_2} = (k_1 + k_2) \cdot \left(1 + 1.4 \cdot 10^{-21} \cdot [H_2O] \cdot \exp\left(\frac{2200}{T}\right)\right) \tag{5}$$

$$k_1 = 2.2 \cdot 10^{-13} \cdot \exp(\frac{600}{T}) \frac{cm^3}{molec\,s} \tag{6}$$

$$k_2 = 1.9 \cdot 10^{-33} \cdot [N_2] \cdot \exp(\frac{980}{T}) \frac{cm^3}{molec\,s} \tag{7}$$

$$[H_2O] = \frac{p^0_{H_2O} \cdot RH \cdot N_A}{R \cdot T} \tag{8}$$

$$p^0_{H_2O}(T) = 1013.25\ hPa \cdot \exp(13.3185a - 1.97a^2 - 0.6445a^3 - 0.1299a^4) \tag{9}$$

$$a = 1 - \frac{373.15\ K}{T} \tag{10}$$

During night, the photochemical production and loss reactions due to OH can be neglected, therefore, the decay of $H_2O_2$ and HCHO in clean air masses during night is dominated by deposition. With the assumption of a constant, horizontally homogenous boundary layer and a linear concentration profile within the BL, the exponential decay can be used to estimate the deposition velocity ($V_{Dep}$) with the method of Shepson et al. (1992). In this calculation, we assume that the initial mixing





ratio $[X]_0$ ($j_{NO2} < 10^{-3}$) represents the mixing ratio on top of the nocturnal boundary layer ($h_{BL}$). If species X follows exponential

decay during night, the first order decay is given by Eq. 11. Thus, the first order decay plot ($\ln[X]_t/[X]_0$ versus the time) yields

the deposition velocity of species X with a known boundary layer height.

$$\ln \frac{[X]_t}{[X]_0} = \frac{-2V_{Dep}(X)}{h_{BL}} t \qquad (11)$$

Additionally, the deposition rate can be calculated with Eq. 12, assuming that the boundary layer is well mixed.

$$V_{Dep} = k_{Dep} h_{BL} \qquad (12)$$

**3 Results**

**3.1 Regional distribution of HCHO, $H_2O_2$ and organic peroxides around the Arabian Peninsula**

The cruise track of the *Kommandor Iona* is shown in Fig. 1, subdivided into eight regions identified by different colors: The Mediterranean Sea (MS), Suez Canal and the Gulf of Suez (SU), Red Sea North (RN), Red Sea South (RS), Gulf of Aden (GA), Arabian Sea (AS), Gulf of Oman (GO) and Arabian Gulf (AG). The AG (also known as the Persian Gulf) and the SU

are well known for their oil and gas industry and intensive ship traffic, respectively, hence primary emissions of $NO_x$, CO, and to a lesser extent HCHO were expected to affect mixing ratios of these trace gases. Nearby ship plumes (including the *Kommandor Iona* exhaust) and other point sources were identified with the use of $NO_x$, CO, $SO_2$ and wind direction data and excluded from the dataset used in this study (Celik et al., 2020). The measurements were affected by aft winds in particular during the first leg, resulting in a contamination from the ships exhaust, thus limiting the amount of data available (Fig. S1).

In general, we did not find elevated mixing ratios of peroxides in ship plumes (presumably a result of the high $NO_x$ levels) and thus decided to keep the data, but corrections for known NO interferences were applied (2.4). Contrary to $H_2O_2$, HCHO is affected by ship exhaust plumes (Celik et al., 2020) and thus contaminated data have been filtered out.



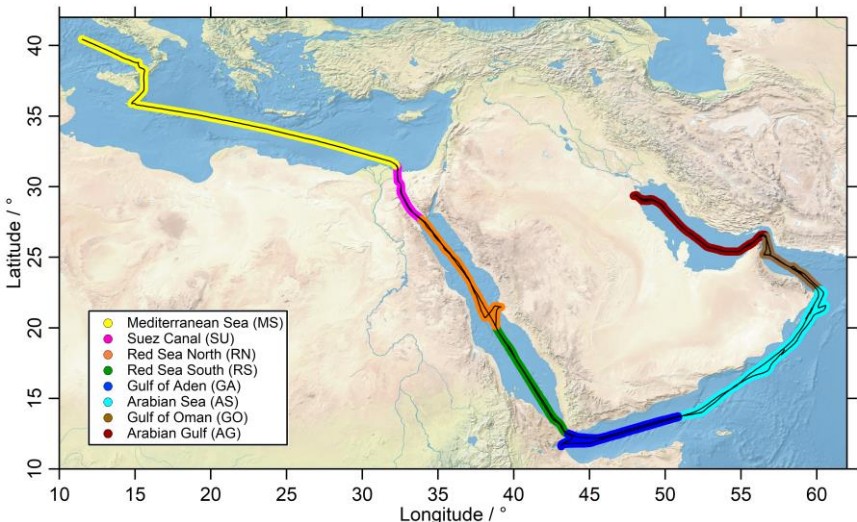


**Figure 1: The shiptrack (black) of the Kommandor Iona during the AQABA cruise subdivided into eight regions: Mediterranean sea (MS: yellow), Suez Canal (SU: pink), Read Sea North (RN: orange), Read Sea South (RS: green), Gulf of Aden (GA: blue), Arabian Sea (AS: turquoise), Gulf of Oman (GO: brown) and Arabian Gulf (AG: red). The map was created with data provided by the Natural Earth website (http://www.naturalearthdata.com).**

In Figure 2 we present mixing ratios of HCHO (upper panels), $H_2O_2$ (middle panels) and ROOH (lower panels) color-coded along the ship cruise track for the first (left panels) and second leg (right panels). Time series of these species can also be found in the supplementary information (Fig. S1, Fig. S3 and S5). Box-and-whisker plots of the mixing ratios for HCHO, $H_2O_2$ and ROOH for the 8 regions are shown in Fig. 3 and 4. Numerical values are listed in Tab. S1.

Mixing ratios of HCHO (upper panels of Fig. 2, Fig. S1) exhibit a large variability. The highest mixing ratios (12.62 ppbv)
were measured in the center of the Arabian Gulf with north-westerly winds originating from Iraq / Kuwait during the first leg (Fig. 2, upper left panel). Lower mixing ratios were detected in this area during the second leg, when the wind originated from the north-east, coming from Iran (Fig. 2, upper right panel). The lowest HCHO median mixing ratios were measured in the RS (0.37 ppbv) during the second leg, in unpolluted air mass originating from Eritrea. Low HCHO was also found in the GA (0.50 ppbv), the MS (0.77 ppbv) and the AS (0.86 ppbv). In general, low mixing ratios of HCHO are associated with low $NO_x$,
low $O_3$ (Tadic et al., 2021), low VOCs (Bourtsoukidis et al., 2019), low OH reactivity (Pfannerstill et al., 2019) and in particular low OH reactivity towards VOCs (Dienhart et al., 2021), while high mixing ratios of HCHO are associated with elevated values for these species.

To the best of our knowledge there are no ship-borne measurements of HCHO available in the Red Sea and the Arabian Gulf to be compared to our data. In general, the measured mean mixing ratios during AQABA are generally larger compared to
previous studies in the MBL. Wagner et al. (2001) performed ship-borne measurements during the INDOEX campaign in the central Indian Ocean with HCHO mixing ratios between 0.2 – 0.5 ppbv, with the lowest mixing ratios in the clean maritime





background of the southern hemisphere. Weller et al. (2000) reported ship-based HCHO measurements in the Atlantic, which reached a broad maximum with values of 1.0 – 1.2 ppbv in the tropical Atlantic.

The lowest median mixing ratios of $H_2O_2$ were found in GO and GA (0.12 ppbv), followed by AS (0.15 ppbv), while higher

mixing ratios of $H_2O_2$ were found in RS, SU (0.25 ppbv) and MS (0.26 ppbv). Altogether, the $H_2O_2$ measurements exhibit lower variation around the Arabian Peninsula compared to HCHO. Higher variability of $H_2O_2$ was found in SU and AG, although less than 25% of the data exceeds 0.50 ppbv, with highest $H_2O_2$ mixing ratios observed in AG (0.92 ppbv) in the harbor of Kuwait. AG, SU, MS and RN also show the strongest diurnal variations of up to ~300 pptv (Fig. S3).

Absolute mixing ratios of $H_2O_2$ are in the same range as previous measurements of $H_2O_2$ in the MBL (Fischer et al., 2015;

Stickler et al., 2007; O'Sullivan et al., 2004; Chang et al., 2004; Kieber et al., 2001; Lee et al., 2000; Weller et al., 2000; Junkermann and Stockwell, 1999). These observations indicate highest mixing ratios (> 500 pptv) of $H_2O_2$ in the tropics (O'Sullivan et al., 2004; Weller et al., 2000; O'Sullivan et al., 1999; Junkermann and Stockwell, 1999; Heikes et al., 1996; Slemr and Tremmel, 1994) and decreasing concentrations towards higher latitudes in both hemispheres, reaching 200 – 300 pptv in the extra-tropics (Fischer et al., 2015; O'Sullivan et al., 2004; Weller et al., 2000; Junkermann and Stockwell, 1999;

O'Sullivan et al., 1999; Slemr and Tremmel, 1994). In general, higher $H_2O_2$ mixing ratios have been observed in continental outflow (e.g. Heikes et al., 1996).

**Figure 2: Overview and data coverage of HCHO, H₂O₂ and organic hydroperoxide measurements during both legs of the AQABA ship campaign (graphs on the left represent the first leg). Contaminated HCHO data (e.g. by ship exhausts) was removed from the dataset with a stack filter (based on the NO, CO and SO₂ observations), therefore there is less HCHO data coverage during the first leg in the Arabian Sea.**





The organic peroxides showed higher variability compared to $H_2O_2$ (Fig. 4), with the lowest median value in AS (0.06 ppbv), followed by GO (0.07 ppbv) and GA (0.10 ppbv). We found the lowest variability (whisker-intervals) in AS, which represents

the cleanest conditions and the lowest variability of $O_3$ and $NO_x$ (Tadic et al., 2020). Higher levels of organic peroxides were detected in SU (0.26 ppbv), AG (0.23 ppbv), MS (0.22 ppbv) and RN (0.20 ppbv) with the highest mixing ratios in the center of the Arabian Gulf during the first leg (2.26 ppbv).

The chromatograms of the HPLC-based instrument indicate significant abundances of four distinct inorganic and organic hydroperoxides in AG (Fig. S10), which were identified as $H_2O_2$, MHP, PAA (peracetic acid) and EHP (ethyl hydroperoxide)

based on their retention times and gaseous injections of PAA with a diffusion source. In addition to the continuous HPLC measurements, we also injected enriched samples with sampling times varying between 12 – 36 h during various times along the ship track. Although these samples have very limited time resolution, they were used for a qualitative assessment of the abundance of further organic hydroperoxides. Significantly enhanced amounts of EHP were only detected over the Arabian Gulf, although small amounts of EHP were also detected in the enriched samples in MS and RS (Fig. S10).

Highest amounts of photochemical air pollution were detected over the AG, which is confirmed by the highest mixing ratios of HCHO and ROOH in this region when observing winds from the western coastline and Kuwait. Less air pollution was observed during the second leg, when we were sampling air masses originating from Iran. In this region we also observed the strongest radiation and the highest temperatures during AQABA. SU and RN also show enhanced contributions of VOCs (Wang et al., 2020) and elevated OH reactivity (Pfannerstill et al., 2019), mainly while passing oil rigs and on the way through

the Suez Canal. Tadic et al. (2020) found the cleanest conditions, from both a $NO_x$ and $O_3$ perspective, in the AS and the RS. This can be confirmed by the rather decreased mixing ratios of HCHO, ROOH and $H_2O_2$ mixing ratios, which reflect low levels of $HO_x$. Air masses from Eritrea also contained suppressed mixing ratios of ROOH during the first leg.

### 3.2 EMAC model comparison

The observations of HCHO, $H_2O_2$ and ROOH were compared to numerical results of the model EMAC. The highly complex

photochemistry around the Arabian Peninsula is well suited to MOM's photochemistry mechanism. The high pollution levels e.g. in the AG contrast with the rather clean regions like AS and MS, that represent mostly aged air masses with less anthropogenic influence, although clean MBL conditions ($NO_x$ < 50 pptv) were rarely encountered during AQABA. Here we use simulations from the lowest vertical level of EMAC (~ 30 m) at a temporal resolution of 10 minutes. Time series (Fig. S1, S3 and S5) and scatter plots (Fig. S2, S4 and S6) are shown in the supplement. Regional variations are highlighted in Fig. 3

and 4 by box-and-whisker plots and by the measurement to model ratio (Fig. 3 and 4), the EMAC data was adapted to the measurements with a time resolution of 10 minutes for these plots. Numerical values are listed in Table S1. Additionally, frequency distribution of mixing ratios for observations and model simulations for the individual regions are presented in the supplement (Fig. S14, S15, S16).

In general, EMAC reproduces the regional trends of HCHO quite well (Fig. 3, left panel) although the whisker intervals

(10 – 90%) demonstrate that the model computes less variability, most likely due to missing localized pollution events, which





is to be expected for a global model at relatively low resolution. The median values of the measured to model ratio vary between 0.3 and 1.6 (Fig. 3, right panel). EMAC overestimates HCHO in the cleaner regions MS, RS and GA, while it underestimates HCHO in AS and the heavily polluted AG.

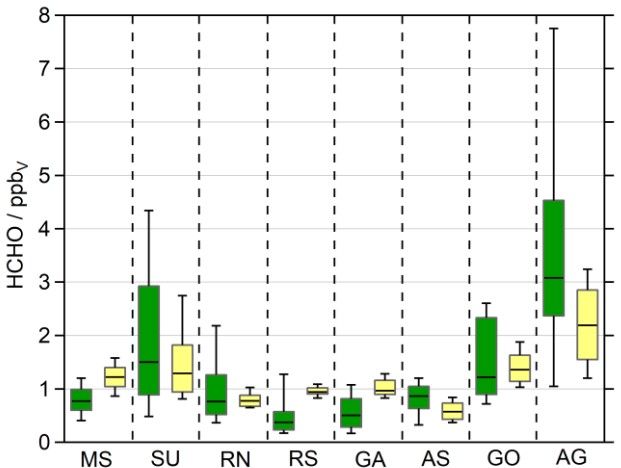
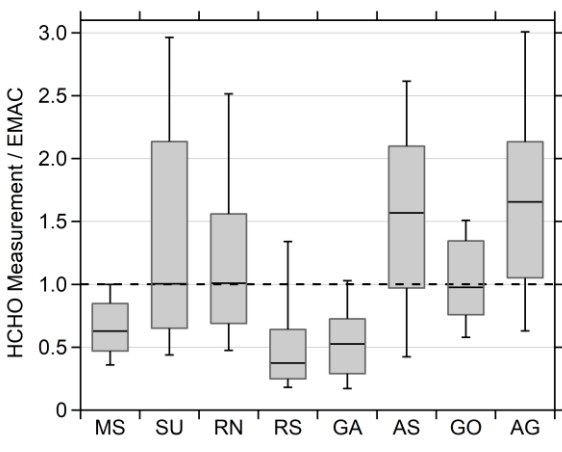

**Figure 3: Formaldehyde observations (green) and EMAC simulations (yellow) divided into the eight regions during AQABA:**
**Mediterranean sea (MS), Suez Canal (SU), Read Sea North (RN), Read Sea South (RS), Gulf of Aden (GA), Arabian Sea (AS), Gulf of Oman (GO) and Arabian Gulf (AG). We used the lowermost model results (~ 30 m) for the comparison in 10 min averages (left). The box represents 25 to 75% of the data and the whiskers 10 to 90% with the median as the black line. The right panel shows the ratio between the observations and the model simulations.**

Elevated $NO_x$ and $O_3$ measurements classified AG and SU as the most polluted regions, followed by RN and GO which both
were influenced to a higher extent by anthropogenic pollution (Tadic et al., 2020). The elevated HCHO during the high ozone and VOC conditions on the first leg in AG was not reproduced by EMAC. Here the model clearly underestimates HCHO (Fig. S1) and ROOH (Fig. S5). Since the elevated OVOCs correlated well with CO and $O_3$ (Wang et al. 2020), we can assume that we probed a highly polluted and photochemically active air mass, with both effective photochemical production and primary emissions of HCHO. During this event, maximum HCHO and ROOH mixing ratios were measured during AQABA, with
values up to 12.63 ppbv HCHO and 2.26 ppbv ROOH in the center of the Gulf (Fig. 2, Fig. S9). The event was less pronounced in EMAC with simulated values reaching up to 3.31 ppbv HCHO leading to the model underestimating HCHO in the AG by about a factor of 4. EMAC does not simulate significantly elevated values of ROOH peaking at 0.49 ppbv, but with elevated contribution of PAA and EHP. Even tough, the model underestimates ROOH also by about a factor of 4 (Fig. S9). Wang et al. (2020) also show, that EMAC simulates enhanced acetone and methyl ethyl ketone (MEK) during this event, although the
model shows no significant increase of acetaldehyde.

In the Suez Canal and the Gulf of Suez (SU), the second most polluted region, the model simulations clearly underestimate the high HCHO mixing ratios encountered during the first leg, even though EMAC also simulates a significant increase. This


is most likely due to local emissions from oil rigs, the dense ship traffic and air masses influenced by biomass burning in this region (Wang et al., 2020).

EMAC overestimates HCHO in the less polluted regions of the MS, RS and GA. Tadic et al. (2020) also found that the simulations overestimate $NO_x$ and $O_3$ in these areas. EMAC significantly underestimates HCHO for the AS, especially during the night (Fig. S1). Here, a well pronounced diurnal cycle is simulated, while the observations indicate only a distinct diurnal variation in the eastern part of AS, which got stronger in GA (Fig. S1). Slightly elevated mixing ratios were observed in AS, compared to clean MBL conditions e.g. during INDOEX (Wagner et al., 2001), and are most likely caused by other primary
sources and oxidation of further VOCs, not by methane oxidation only. Previous observations in the remote MBL showed significantly lower HCHO in the range of 0.2 – 0.4 ppbv (Wagner et al. 2001). Thus, we assume that the air masses encountered in the AS during AQABA were still influenced by anthropogenic pollution, which is supported by the elevated $NO_x$ (Tadic et al., 2020) and acrolein as the main contributor to OH reactivity in AS and GA (Pfannerstill et al., 2019). An additional source for HCHO was the ozonolysis of ethene, which reached maximum values of 0.24 ppbv with a median of 0.08 ppbv in AS
(Bourtsoukidis et al., 2019). Wang et al. (2020) also show strongly enhanced acetaldehyde in AS relative to model simulations, indicating a missing oceanic source in the model. Tripathi et al. (2020) also performed VOC and sea water measurements of phytoplankton species, which demonstrate the high biological activity in the region. They determined elevated ethene (8.92 ± 3.50 ppbv) and propene (3.38 ± 1.30 ppbv) in marine air originating from the Arabian Sea. Just recently, Tegtmeier et al. (2022) highlighted the complexity of the air phase composition over the Indian Ocean, with the major differences between
the Indian summer and winter monsoon.

Altogether, EMAC reproduces observed HCHO mixing ratios on average within a factor of two. However, the model clearly underestimates the local maxima, especially for the AG, which leads to the assumption of missing sources in EMAC. The model simulates the AS cleaner than it was observed, but the relatively clean RS (with winds from Eritrea) was overestimated. Given the multitude of potential HCHO sources, both from direct emissions and a large variety of photochemical precursors,
the agreement is quite good, although the observations highlight that VOC emissions in the AG need to be adjusted.





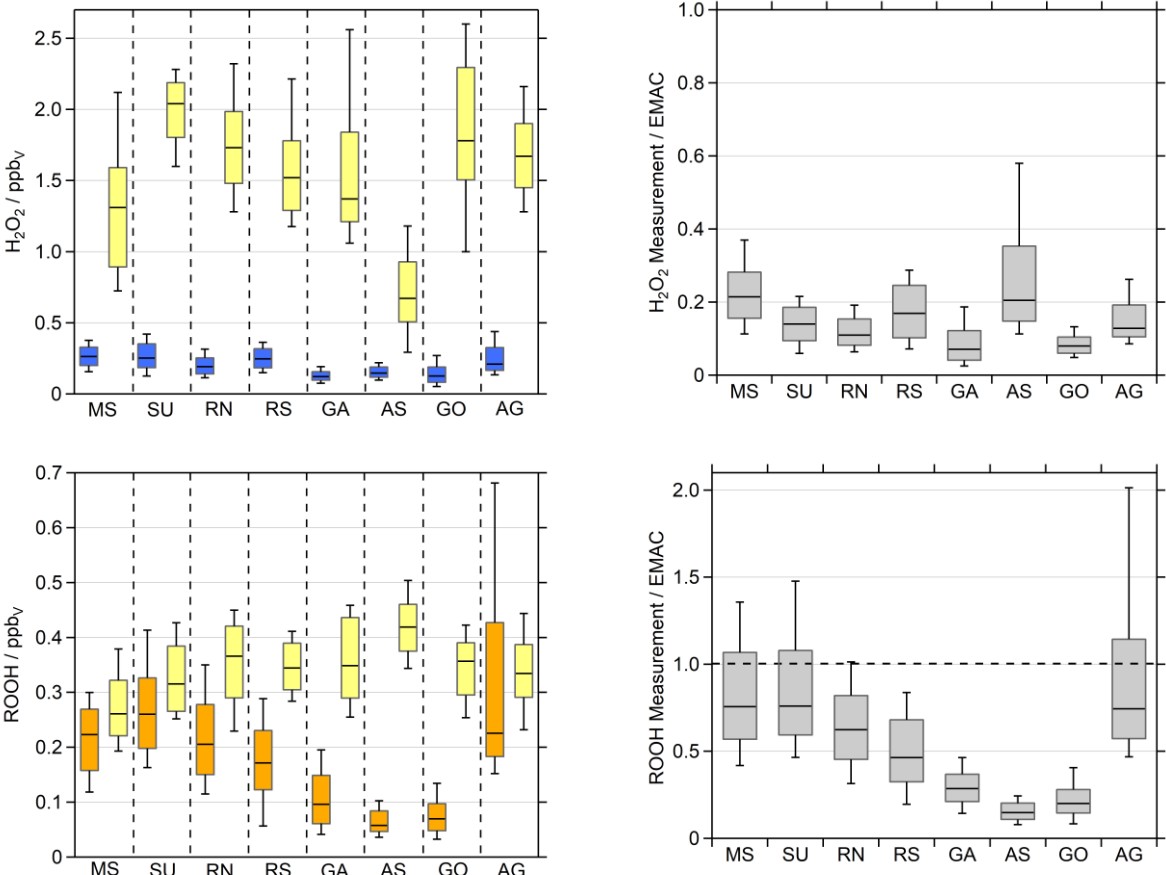

**Figure 4: H₂O₂ (blue) and organic hydroperoxide (ROOH, orange) observations compared to the EMAC simulations (yellow), we used the lowermost model results (~ 30 m) for the comparison. The box represents 25 to 75% of the data and the whiskers 10 to 90% with the median as the black line. The right panel shows the ratio between the observations and the simulations.**

The model-measurement comparison for H₂O₂ is even worse as EMAC systematically overestimates H₂O₂ mixing ratios by up to an order of magnitude. With the exception of the Arabian Sea (0.7 ppbv), the model predicts H₂O₂ with median mixing ratios in excess of 1.3 ppbv, with the highest median values of 2.0 ppbv and 1.8 ppbv for SU and GO, respectively. While the measurements cover a whisker range (10 to 90 % of the data) of only 0.1 to 0.4 ppbv. This consistent overestimation by the model indicates either a significant overestimation of H₂O₂ sources or missing sinks in the model, or a combination of both.

In order to compare the observations of ROOH with model results, we summed up individual simulated organic hydroperoxide species, which were identified in the qualitative HPLC measurements (Fig. S4): methyl hydroperoxide (MHP), peracetic acid (PAA) and ethyl hydroperoxide (EHP). Please note that measured ROOH is a lower limit of the sum of organic hydroperoxides, since different sampling efficiencies for the individual species, which depend on the Henry's law constants, are not accounted





for. It can be assumed, that MHP is the dominant contributor to the total organic hydroperoxides in the clean MBL, which has a sampling efficiency of only 60% (Fischer et al., 2015). In remote areas, this would lead to an underestimation of measured ROOH by a factor 0.6. With significant contributions of higher organic hydroperoxides, which are generally more soluble, this underestimation tends to be smaller.

EMAC also tends to overestimate the organic hydroperoxides, with the lowest median value of 0.26 ppbv in MS (observations

0.22 ppbv), and the highest of 0.42 ppbv in AS (observations 0.06 ppbv) (Fig. 4 and Table S1). For the whole dataset, the simulated ROOH cover a whisker range (10 to 90 % of the data) between 0.19 to 0.50 ppbv, while the observations yield a span of 0.04 to 0.68 ppbv. Although the measurements can be reproduced within the 25 to 75 % box range in some regions, median values differ between about a factor of 1 to 7 between the simulations and observations. Please note that AG was the only region where we measured four separated hydroperoxides in the in situ results of the HPLC, with the largest contribution

of MHP and EHP. This enhancement was also found in the EMAC results, as EHP and PAA mixing ratios increased in AG, especially during the high pollution events of the first leg (Fig. S10). Although the simulations of ROOH match the observations better than $H_2O_2$, EMAC overestimates the organic peroxides, especially in the clean regions while cruising close to the coast. AG shows the highest variability of ROOH, which is to be expected due to the complex photochemistry of VOCs (Bourtsoukidis et al., 2019; Pfannerstill et al., 2019). The model simulates strong diel cycles (~300 pptv) throughout the whole

dataset, while the observations only indicate comparable variations for the RN, SU, MS and AG. A distinct decline in ROOH mixing ratios was observed for the AS and GO, which is not reproduced by EMAC.

Possible explanations for the systematic overestimation of both $H_2O_2$ and organic hydroperoxides by EMAC can be an overestimation of photochemical sources or an underestimation of loss processes in the model, or due to the unknown sampling efficiencies of the organic peroxides. To investigate photochemical misrepresentations, we compared the observed and

simulated OH and $HO_2$ daytime values ($j_{NO2} \geq 10^{-3}$ s$^{-1}$; Fig. 5, S7, S8). Since the source term of $H_2O_2$ depends quadratically on $HO_2$ concentrations (Eq. 3), simulations of $H_2O_2$ are highly sensitive to $HO_2$, while its photochemical loss scales linearly with OH (Eq. 4). The model overestimates both OH and $HO_2$ throughout the whole campaign. Highest observed $HO_2$ median values were found in SU (19.3 pptv) and RN (16.3 pptv), followed by RS (14.8 pptv) and MS (14.4 pptv). The remaining regions show significantly less $HO_2$ with the smallest median value for GO (4.5 pptv). The smallest whisker ranges of the dataset in

AS and GO demonstrate suppressed $HO_2$ in these regions. Surprisingly small mixing ratios were also detected over the polluted Arabian Gulf (6.8 pptv).

According to the observations, EMAC simulates the highest $HO_2$ median values for SU (27.9 pptv) and RN (26.1 pptv), while mixing ratios in MS (19.4 pptv) and RS (11.2 pptv) are smaller. Significant overestimation of $HO_2$ was found for GO (23.1 pptv) and AG (19.0 pptv) show similar enhancements of $HO_2$. In the rather clean regions during AQABA, e.g. GA

(16.7 pptv) and AS (19.6 pptv), the model generates significantly enhanced daytime $HO_2$ compared to the observations, while diurnal variation of $HO_2$ matches the observations in RS and MS (Fig. S8). Altogether, we examined an overestimation of $HO_2$ by about a factor of 2 (EMAC dataset was aligned to the observations), with average daytime mixing ratios of 11.3 pptv for the observations and 19.7 pptv for the EMAC simulation, respectively.





Highest median OH values were observed for SU (0.13 pptv), RN (0.13 pptv), MS (0.12 pptv) and GO (0.11 pptv). Slightly
less OH was detected in RS (0.07 pptv), AS (0.05 pptv) and AG (0.05 pptv). The box and whisker ranges indicate highest
variations of OH in GA, where we detected the highest OH mixing ratios of 0.6 pptv close to Bab-el-Mandeb on the 16.18.2017
(Fig. S7). EMAC simulates the highest OH mixing ratios in SU (0.49 pptv), GO (0.47 pptv) and RN (0.44 pptv), while the
lowest median values of OH were simulated in RS (0.13 pptv), followed by MS (0.26 pptv) and AS (0.26 pptv).

The EMAC results for OH follow similar regional trends compared to $HO_2$, although they reflect a stronger pronounced
overestimation, since the measurements do not reflect a substantial increase of OH in MS, SU, RN and RS. This results in
overestimated daily median values within a factor of 2 to 5.

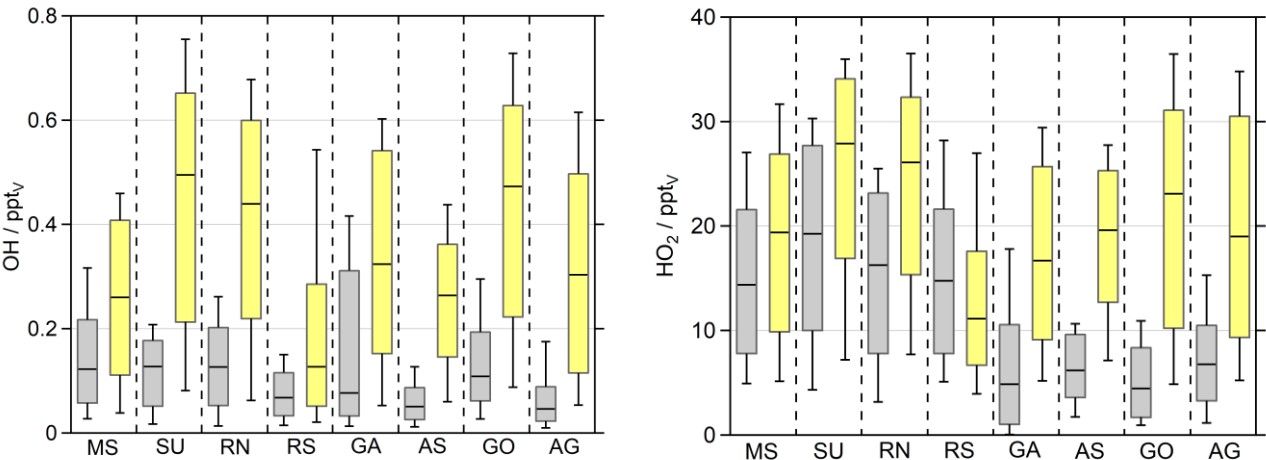

**Figure 5: OH and $HO_2$ daytime values ($j_{NO2} \geq 10^{-3}$ s$^{-1}$) of the observations (grey) and the EMAC simulations (yellow). The box represents 25 to 75% of the data and the whiskers 10 to 90% with the median as the black line. The EMAC data was adapted to the measurements with a time resolution of 10 minutes, so that the diurnal variations are reflected accurately.**

**3.3 Photochemical production and loss of $H_2O_2$**

The comparison to EMAC in the previous section showed that overestimations of $HO_x$ by EMAC affect simulations of HCHO,
$H_2O_2$ and ROOH. How mis-representation of $HO_x$ will affect HCHO is complex due to the many HCHO sources and the fact
that both sources and sinks are strongly related to OH concentrations. Therefore, we will concentrate on $H_2O_2$ in the following
calculation of its photochemical production and loss terms in order to evaluate the discrepancy between modelled and measured
$H_2O_2$. $H_2O_2$ is highly sensitive towards deviations of $HO_x$ between the observations and the model, as its production depends
quadratically on $HO_2$ (Eq. 3), but its loss only linearly on OH (Eq. 4). Thus, $H_2O_2$ can be used to evaluate the discrepancy
between measured and modelled $HO_x$.

The daytime production rates of $H_2O_2$ ($j_{NO2} \geq 10^{-3}$ s$^{-1}$) are displayed in Fig. 6 in order to compare the results of the observations
and EMAC, with the corresponding timelines presented in Fig. S11 and a scatter plot of $k_{HO2+HO2}$ in Fig. S12. Based on the



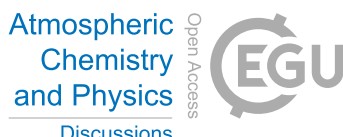
observations, the highest production rates were found in SU with a median production of 202 pptv h$^{-1}$ and the largest whisker
range of up to ~480 pptv h$^{-1}$. A similar range was obtained by EMAC, although with an increased median value of 298 pptv h$^{-1}$. MS and RN demonstrate comparable results, where the box ranges of the observations and the model agree to some extent, although the overestimated HO$_2$ by EMAC outweighs the slightly smaller reaction constant of $k_{HO2+HO2}$ (Fig. S12). Overall, EMAC tends to overestimate P(H$_2$O$_2$), except in RS, where the model correctly simulates the lowest H$_2$O$_2$ production rates of the dataset with a median value of 63 pptv h$^{-1}$ – about a factor of 5 lower than in SU. However, the observations display

reduced P(H$_2$O$_2$) in the remaining regions, which resulted in a stronger pronounced discrepancy with at least a factor of 5 for GA, AS, GO and AG.

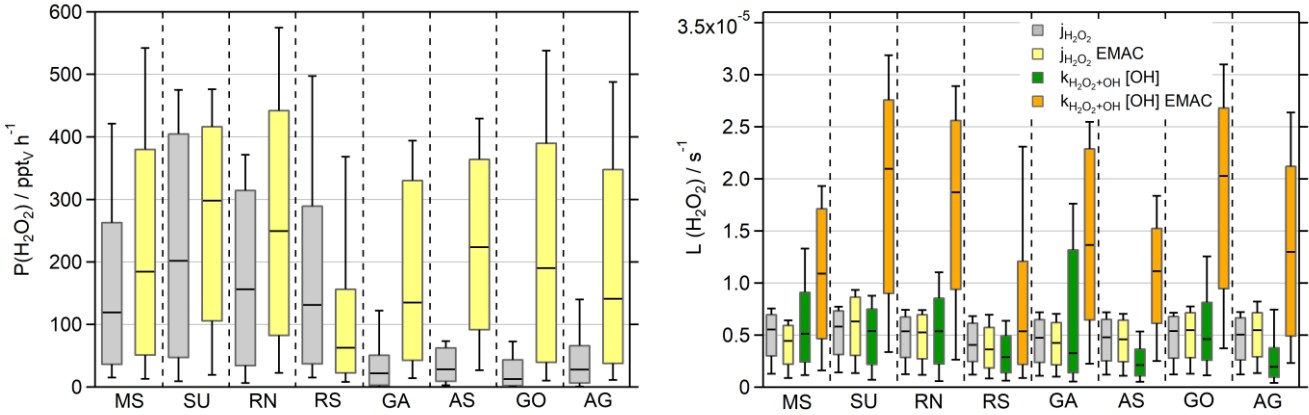

**Figure 6: Box plots of H$_2$O$_2$ production rates (pptv h$^{-1}$) for the observations (grey) and the EMAC model (yellow) during daytime ($j_{NO2} \geq 10^{-3}\,s^{-1}$) and the corresponding loss rate constants (s$^{-1}$) for photolysis (observations in grey and EMAC in yellow) and reaction**
**with OH (green, orange). The boxes represent 25 to 75% of the data and the whisker intervals 10 to 90% with the median values as the black lines. The corresponding timelines are presented in Fig. S11.**

Photochemical losses of H$_2$O$_2$ are the reaction with OH and photolysis, which were calculated according to Eq. 4, and displayed without multiplication of the H$_2$O$_2$ mixing ratio for the sake of comparability (Fig. 6). The simulated photolysis rate constants (J-values) demonstrate good agreement with the observations with a factor of 1.2, during the rarely cloudy conditions of

AQABA. EMAC only overestimates $j_{H2O2}$ for air masses very close to the coastline, e.g. in SU (up to ~1.5 · 10$^{-6}$ s$^{-1}$) and to a lesser extent in GO, while the photolysis rate was underestimated in RS (Fig. S11). Loss of H$_2$O$_2$ due to photolysis was less important than the reaction with OH for most regions, while photolysis prevailed for AS, RS and AG. In contrast to the observations, photochemical losses of H$_2$O$_2$ were dominated by the reaction with OH and were overestimated within a factor of ~2 – 5 by the model with the best agreement in RS.

To better put these results into perspective, Fig. 7 presents the net photochemical production of H$_2$O$_2$ (P(H$_2$O$_2$) – L(H$_2$O$_2$)), whereby the loss rate constants were multiplied with H$_2$O$_2$ mixing ratio. Please note that in this term neither physical processes nor transport are represented and thus it only reflects the effect of photochemistry on the H$_2$O$_2$ mixing ratio. The slightly





overestimated photochemical production in MS, SU and RN by EMAC is compensated by elevated losses via reaction with OH in these regions, so that both datasets agree well in MS and RN, and demonstrate a strongly pronounced diurnal variation

peaking at ~580 pptv h$^{-1}$ and ~500 pptv h$^{-1}$, respectively (Fig. 8). Net photochemical production of $H_2O_2$ outweighs the model results in SU and RS, especially during noon (Fig. 8). GA, AS, GO and AG remain overestimated by EMAC, so that the elevated losses in the model do not compensate for the enhanced production rates due to the quadratic dependence on the $HO_2$ concentration. The observations demonstrate less net photochemical production in GA, AS, GO and AG with noontime values below 200 pptv h$^{-1}$ due to the decreased $HO_2$. Pfannerstill et al. (2019) reported highest OH reactivity in AG (11.6 s$^{-1}$) and SU

(10.4 s$^{-1}$ – 10.8 s$^{-1}$), comparable results for GO (8.4 s$^{-1}$) and GA (8.0 s$^{-1}$); and the lowest OH reactivity for AS (4.9 s$^{-1}$). Air masses in AG demonstrated by far the highest contribution of reactivity towards OVOCs (~40%), alkanes and alkenes (together ~14%). Air masses in GO showed slightly higher contributions of reactions with $NO_x$ compared to AG, while AS represents the cleanest conditions during AQABA with respect to $NO_x$ (Tadic et al., 2020). A potential explanation for the surprisingly low $HO_2$ mixing ratios in AG could be suppressed OH recycling by means of enhanced organic peroxy radicals ($RO_2$) and in

general high contribution of reactions with OVOCs, alkanes, alkenes and aromatics, as OH recycling through these reactions is slower compared to $NO_x$ recycling (via $HO_2$ + NO). Enhanced ROOH indicate a higher contribution of $RO_x$ reactions (= OH + $HO_2$ + $RO_2$), which would also slow down OH recycling. There are no measurements of organic peroxy radicals available, but Tadic et al. (2020) calculated noontime estimates of $HO_2$ + $RO_2$, with the highest noontime median values of ~75 pptv in AG (see Tadic et al., 2020 Fig. 7). The remaining regions of AQABA show noontime median values in the range

of ~10 – 35 pptv. Elevated $RO_2$ in AG is also supported by the enhanced mixing ratios of ROOH, the only region where we detected MHP, PAA and EHP in the in-situ measurements of the HPLC (Fig. S10).

The decreased observations of net photochemical production of $H_2O_2$ in AS, GO and GA are generally caused by lower $HO_x$ mixing ratios, which are not reproduced by EMAC. The box range indicates similar values of net photochemical production in MS, which agree mostly within a factor of 2. However, the $H_2O_2$ observations display an average diurnal variation of

~0.2 ppbv with highest mean mixing ratios of ~0.4 ppbv at 12 UTC, while the EMAC results indicate a variation of ~1.2 ppbv with highest mixing ratios of ~2.1 ppbv also at 12 UTC (Fig. S13). In general, the observations only display weak diurnal variations compared to the large diurnal variation in net photochemical production, which implies that other loss processes (e.g. deposition, wash out due to sea spray) contributed significantly to the diurnal variability of $H_2O_2$ in the marine boundary layer.






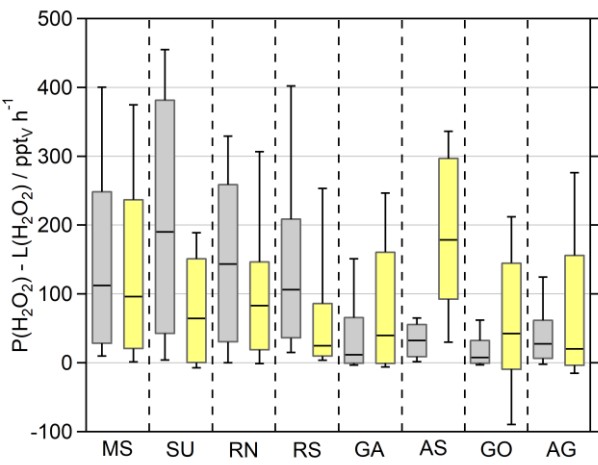

**Figure 7: Box Plot of the net photochemical production of H₂O₂ (P(H₂O₂) – L(H₂O₂)) of the observations and the EMAC model results (yellow) during noon ($j_{NO2} \geq 10^{-3}\,s^{-1}$). The boxes represent 25 to 75% of the data and the whisker intervals 10 to 90% with the median values as the black lines. The corresponding timelines are presented in Fig. S11.**

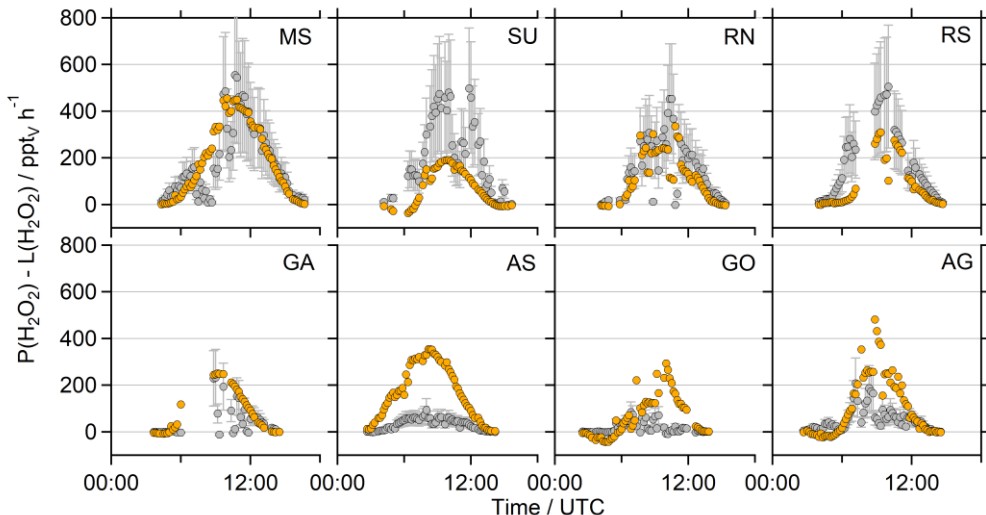

**Figure 8: Diurnal variation of the net photochemical production of H₂O₂ (P(H₂O₂) – L(H₂O₂)) of the observations (grey) and the EMAC model results (orange) during noon ($j_{NO2} \geq 10^{-3}\,s^{-1}$). The corresponding timelines are presented in Fig. S11.**

## 3.4 Dry deposition and transport

An alternative reason for deviations between $H_2O_2$ observations and EMAC predictions could lie in the physical processes of deposition to the ocean surface and entrainment through the top of the MBL. Both processes are related to the absolute value

and the diurnal variability of the boundary layer height. Due to the coarse resolution of EMAC grid cells (approx. 120 km),





the BLH in the model is often affected by diurnal variation due to neighboring continental cells, especially close to the coast. Figure 9 shows EMAC simulations of BLH compared to ERA5 data (ECMWF ReAnalysis 5[th] generation). ERA5 resolves the global atmosphere in hourly intervals for 30 km grids at 137 vertical levels up to 0.01 hPa and thus its horizontal resolution is a factor of 4 higher than that of EMAC. The ERA5 dataset is available within the Copernicus Climate Change Service 590 (https://www.ecmwf.int/en/forecasts/datasets/reanalysis-datasets/era5; last access 27.02.2021). The BL simulated by EMAC is very shallow during night and increases rapidly in height after sunrise, which may reflect continental influence in the EMAC grid-boxes. The only regions where EMAC shows no clear continental influence are the AS and the MS, even though the model clearly underestimated the BLH over the Arabian Sea and even more over the Mediterranean Sea. Local maxima on the 27[th] and 30[th] August are prominent while passing Crete and the Strait of Messina.

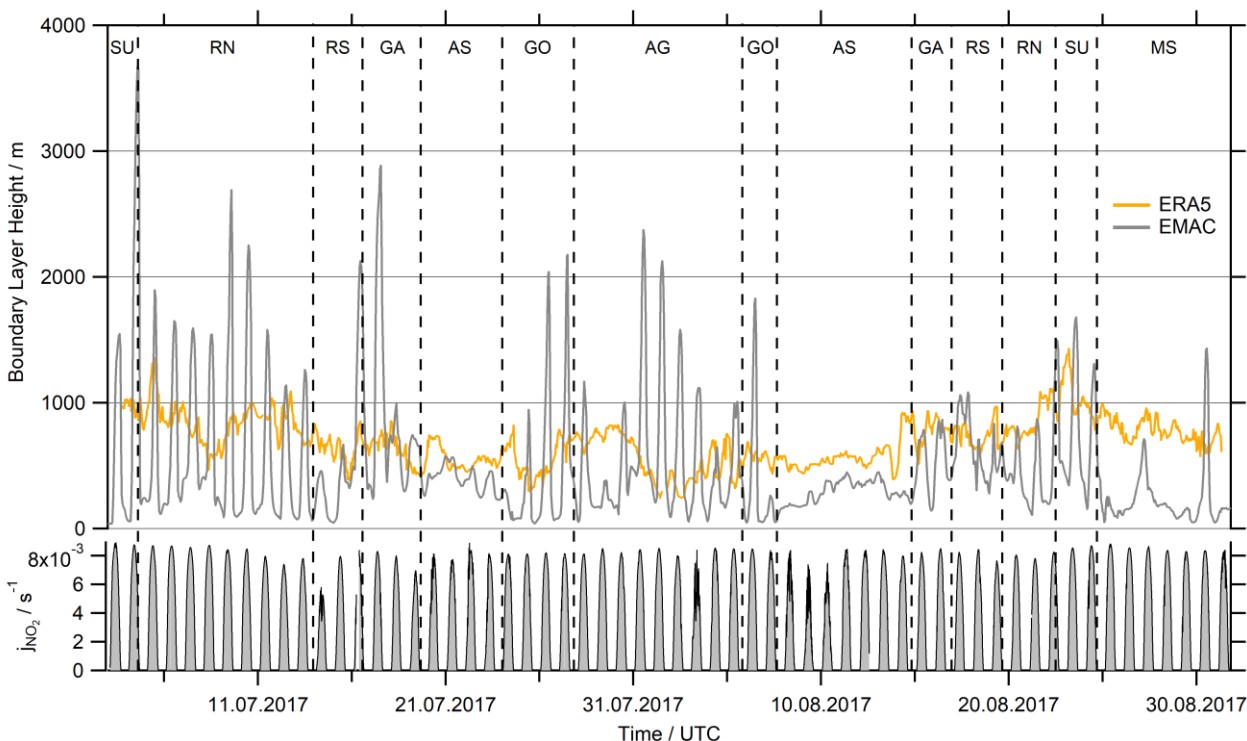


**Figure 9: Comparison of the simulated BLH in EMAC with ERA5 (ECMWF ReAnalysis 5[th] generation) data at a four times higher resolution (31 km grid) than EMAC (~120 km grid) and hourly averaged data. EMACs BLH indicates continental influence by the strong diurnal variation while cruising close to the coast (SU, RN, GA, GO and AG, with the highest values determined for the ports of Jeddah (10. – 13.07.), Djibouti (16.07.) and Kuwait (01. – 03.08.). The measured photolysis frequency $j_{NO2}$ serves as a reference of** 600 **sunlight intensity during AQABA.**

This misrepresentation of the MBL height and its diurnal variation by EMAC has two consequences. First, according to Eq. 12, the deposition loss $k_{Dep}$ for a given deposition velocity $V_{Dep}$ is inversely proportional to the boundary layer height $h_{BL}$. Overestimations of $h_{BL}$ by EMAC in particular during the day would thus lead to an underestimation of the deposition sink,





while it would lead to an overestimation during the night. Additionally, diurnal variations of $h_{BL}$ lead to entrainment of free

tropospheric air into the MBL, in particular during the early morning (Fischer et al., 2015; Fischer et al., 2019). While vertical profiles of HCHO mixing ratios decrease with height (Anderson et al., 2017; Klippel et al., 2011; Heikes et al., 2001), $H_2O_2$ and MHP mixing ratios increase up to a local maximum above the boundary layer (Allen et al., 2022; Nussbaumer et al., 2021a; Klippel et al., 2011). Therefore, intrusion of air masses from the lower troposphere will most likely result in a decrease of HCHO in the MBL, while peroxide mixing ratios would likely increase as shown in Fischer et al. (2015).

To further evaluate the influence of deposition on $H_2O_2$ and HCHO levels, deposition velocities were derived from nighttime observations ($j_{NO2} < 10^{-3}$ s$^{-1}$) of their loss rates following the method of Shepson et al. (1992) (Eq. 13). Here we use the exponential decays of the HCHO and $H_2O_2$ mixing ratios versus time to deduce nighttime loss rates in the Arabian and the Mediterranean Sea, where the EMAC simulation of $h_{BL}$ was most accurate (Fig. 9, 10). The slope of the linear regression yields the respective deposition rate constant ($k_{Dep}$) during night, assuming negligible nighttime chemistry, i.e. non-significant

production of $H_2O_2$ and HCHO due to ozonolysis and nighttime oxidants (NO$_3$, Cl) and their respective losses. We only used the data of the second leg in AS (less data coverage on the first leg due to contamination), which represent high humidity conditions (87.6 ± 3.1 %) and strong headwinds (10.3 ± 2.3 m s$^{-1}$), while winds over the Mediterranean Sea were slower (6.1 ± 2.2 m s$^{-1}$, 73.6 ± 7.0 %). In general, deduced loss rates for $H_2O_2$ and HCHO show higher variability in AS than in MS, with highest values for both species on the 12$^{th}$ of august, and lowest values on the 9$^{th}$ of august in the eastern part of AS after

a partly cloudy day.

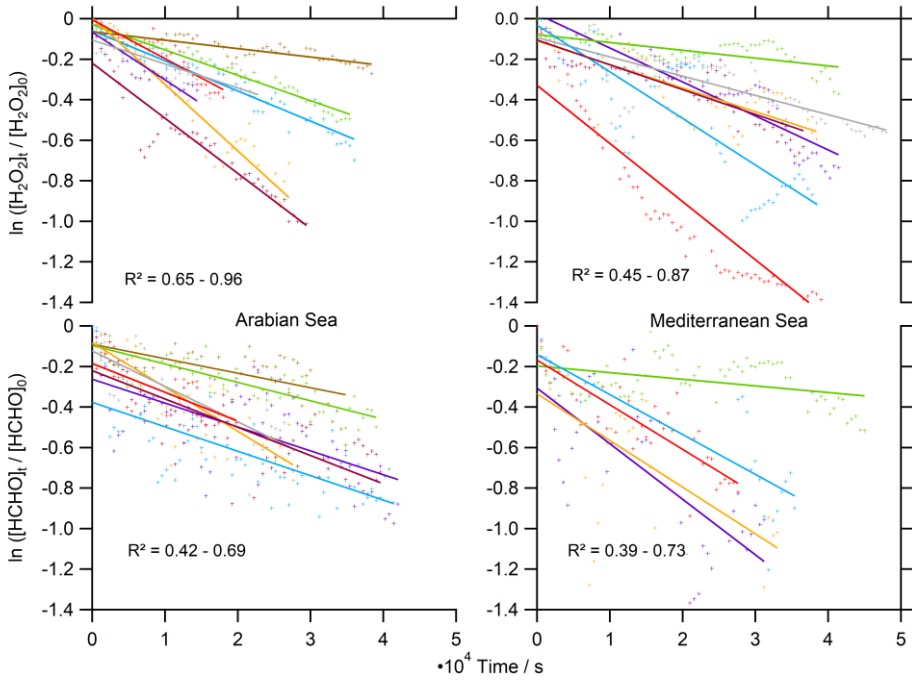

**Figure 10: Determination of the deposition rates $k_{dep}$ for $H_2O_2$ and HCHO in 10 minute averages during night ($j_{NO2} < 10^{-3}$ s$^{-1}$).**





A mean deposition loss $k_{Dep}(H_2O_2)$ of $1.83 \pm 0.93 \cdot 10^{-5}$ s$^{-1}$ was determined for the Arabian Sea, similar to the results in the Mediterranean Sea ($1.51 \pm 0.85 \cdot 10^{-5}$ s$^{-1}$). We thereby determined a minimum of $0.37 \pm 0.15 \cdot 10^{-5}$ s$^{-1}$ and a maximum

deposition loss rate of $2.87 \pm 1.15 \cdot 10^{-5}$ s$^{-1}$ for $H_2O_2$ with a variation of an order of magnitude. Dry deposition rates of HCHO are comparable to $H_2O_2$ with $k_{Dep}(HCHO) = 1.34 \pm 0.46 \cdot 10^{-5}$ s$^{-1}$ in the AS and $1.91 \pm 0.93 \cdot 10^{-5}$ s$^{-1}$ in the MS. The deposition losses of HCHO cover a similar range of $0.33 \pm 0.13 \cdot 10^{-5}$ s$^{-1}$ to $2.74 \pm 1.10 \cdot 10^{-5}$ s$^{-1}$. The results for MS cover less nights, as we experienced local increases of HCHO from the 24.08. to the 26.08.17. These enhancements were associated with elevated $NO_2$, indicating local pollution events and thus we excluded these nights.

Based on the deposition losses we calculated deposition velocities according to Eq. 12 using mean values of $h_{BL}$ from ERA5 data for the corresponding timeframe. For the resulting $V_{Dep}$ we assume an uncertainty of at least 40 % (Fig. 11, Table S2). $V_{Dep}(HCHO)$ cover a range of $0.23 – 2.22$ cm s$^{-1}$, with the highest values during the night of the $27.08. – 28.08.17$. in the Mediterranean Sea. Mean values ($\pm 1\sigma$) of $0.77 \pm 0.29$ and $1.49 \pm 0.76$ were determined for AS and MS, respectively. The deposition velocities of $H_2O_2$ cover a similar range of $0.26 – 2.34$ cm s$^{-1}$, also with the highest values during the night of the

$27.08. – 28.08.17$. This resulted in mean values of $1.03 \pm 0.52$ cm s$^{-1}$ for AS and $1.21 \pm 0.69$ cm s$^{-1}$ for MS. Averaged values were compared to the $V_{Dep}$ used by EMAC in Fig. 11. In general, observation based $V_{Dep}$ and model values for both species are of similar magnitude for AS (with the exception of the very low values derived during the night of the $09.08. – 10.08.17$), while values of $V_{Dep}$ are underestimated by at least a factor of 2 for the Mediterranean Sea. Additionally, EMAC simulates less variability compared to the observations with mean values ($\pm 1\sigma$) of $0.78 \pm 0.16$ cm s$^{-1}$ for AS and

$0.32 \pm 0.16$ cm s$^{-1}$ for MS, respectively. Deposition velocities of $H_2O_2$ show enhanced values compared to HCHO due to the larger henry coefficient of $H_2O_2$, which resulted in mean values of $1.05 \pm 0.27$ cm s$^{-1}$ for AS and $0.37 \pm 0.21$ cm s$^{-1}$ for MS. Striking similarities were found for both species, as the calculated $k_{Dep}$, and also the derived $V_{Dep}$, seem to follow the same trend in the Arabian Sea. A linear correlation coefficient of $R^2=0.77$ was found for a linear fit of $V_{Dep}(HCHO)$ against $V_{Dep}(H_2O_2)$ for the Arabian Sea. The simulated deposition velocity in EMAC depends linearly on the wind speed (Fischer et

al., 2015), which explains the higher values derived during the strong head winds in the Arabian Sea. The observations confirm larger deposition velocities of $H_2O_2$ for the AS, while we determined enhanced values of $V_{Dep}(HCHO)$ for the MS. Both species show declined deposition velocities close to the coast on the 30.07.17, and in general a higher variability than the simulations by EMAC.

The observed nighttime values of $V_{Dep}(H_2O_2)$ match previously derived values within the literature: Allen et al. (2022) found

similar values of $V_{Dep}(H_2O_2)$ in the MBL with a range of $1.00 – 1.32$ cm s$^{-1}$, which corresponds to a loss of $5 – 10\%$ $HO_x$ in the marine boundary layer. Stickler et al. (2007) determined a mean $V_{Dep}(H_2O_2)$ of 1.3 cm s$^{-1}$ in the MBL with a range of 0.1 to 1.8 cm s$^{-1}$ depending on the entrainment rate. Fischer et al. (2019) calculated nighttime deposition velocities in the continental boundary layer for five different campaigns in Europe and determined values in the range of 0.16 to 0.60 cm s$^{-1}$ during night, and 0.56 to 6.04 cm s$^{-1}$ during the day. Nguyen et al. (2015) performed flux measurements and derived a diel

cycle of $V_{Dep}(H_2O_2)$ with values $\leq 1$ cm s$^{-1}$ during night and a maximum of ~6 cm s$^{-1}$ during noon. In sum, our observations agree with previous measurements in the continental and marine boundary layer during night.




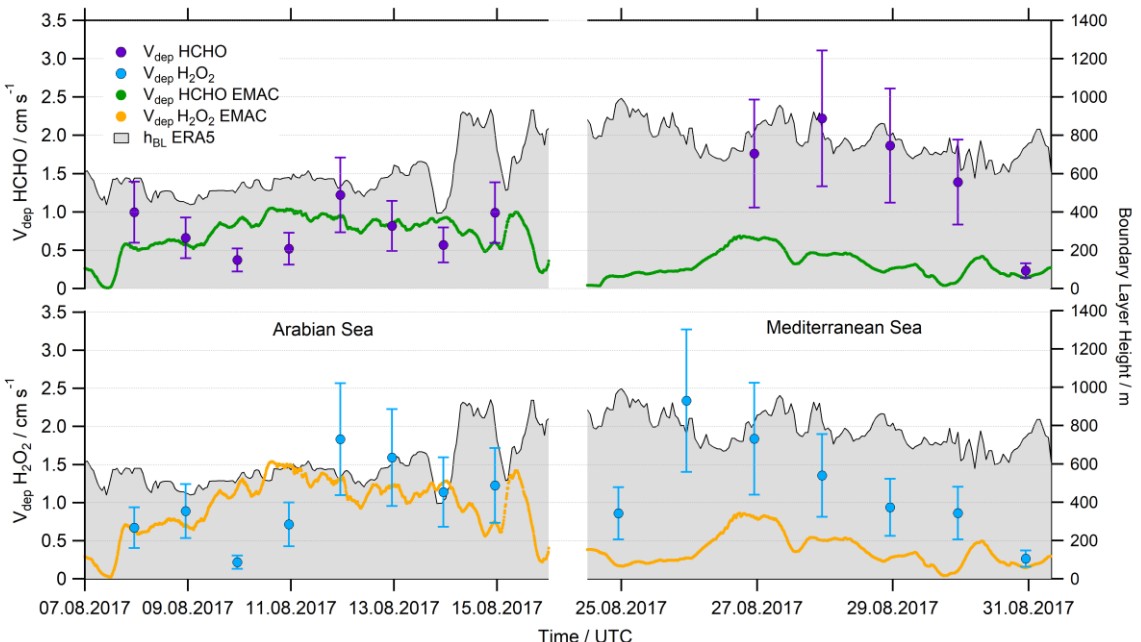

**Figure 11: Comparison of the calculated deposition velocity $V_{dep}$ for HCHO and $H_2O_2$ in the Arabian (AS) and the Mediterranean Sea (MS).**

Deposition velocities of HCHO are generally expected to be lower than that of $H_2O_2$ due to its higher uptake resistance, which is related to e.g. solubility and hydrolysis of the trace gas (Stickler et al., 2007; Ganzeveld and Lelieveld, 1995). This can be confirmed with our results in AS, where $V_{Dep}$(HCHO) is on average a factor of 0.8 smaller than $V_{Dep}$($H_2O_2$). The results in MS do not confirm this expectation, as we determined a factor of 1.2 higher deposition velocities of HCHO. To the best of our knowledge, reports of the deposition velocity of HCHO in the MBL are sparse. Nussbaumer et al. (2021b) derived a nighttime

$V_{Dep}$(HCHO) of 0.47 cm s$^{-1}$ in the continental boundary layer. Sumner et al. (2001) calculated a $V_{Dep}$(HCHO) of $0.65 \pm 0.36$ cm s$^{-1}$ during night at a mixed deciduous/coniferous forest site, while Stickler et al. (2007) suggest a constant value of 0.36 cm s$^{-1}$ over the ocean based on a single-column model. In comparison to the evaluated $V_{Dep}$ in the continental boundary layer, our measurements indicate more efficient deposition over the open ocean. This may be due to a more efficient near-surface transport, e.g. due to high wind speeds and turbulence as it is expected that high wind speeds lead to a more efficient

deposition. Altogether, EMAC simulated accurate dry deposition velocities of HCHO and $H_2O_2$ for the AS with a deviation of less than 5% for the derived mean values, while $V_{Dep}$ was underestimated by at least a factor of 2 for MS. Additionally, the observations demonstrate a higher variability than the model. Please note that deposition velocities for the observations were only calculated during night, and thus deposition losses of $H_2O_2$ and HCHO during daytime remain uncertain, although previous observations indicate a stronger pronounced deposition loss during the day (e.g. Fischer et al. 2019). Please note that

our observations might be affected by additional loss processes, e.g. due to interactions with sea spray or in general heterogeneous chemistry.



on soil moisture, which decreased HCHO mixing ratios during boreal summer by up to 25% on ground level. We want to highlight that the ocean surface might be an additional surface where dry deposition of trace gases might need to be adjusted. This effect may be most important on a local scale, but certainly could have an effect on the $HO_x$ budget in EMAC. Additionally, EMAC is limited by its coarse spatial resolution (~110 km), which leads to spurious diurnal variation of the boundary layer height when cruising close to the coast.

The overestimated $HO_x$ in EMAC most likely results from the overall enhanced VOC oxidation. Additionally, the model was not able to reproduce the local phenomena encountered, e.g. the air pollution event in AG. This leads to the assumption that the model may reproduce HCHO within a factor of 2, but partly due to wrong reasons, i.e. overestimated $HO_x$ which compensates for missing sources in the model. This assumption is supported by the observations of several other OVOCs which were not matched by EMAC (Wang et al., 2020).

The systematic overestimation of $H_2O_2$ is at least partly explained by the overrated $HO_2$, although net photochemical production of $H_2O_2$ revealed that the model matches the observations in some regions well, as the overall overestimated OH compensates partially for too high values of $HO_2$. The decreased observations of $HO_2$ in GA, AS, GO and AG are not matched by EMAC. The reduced $HO_x$ encountered in air masses over the Arabian Gulf can most likely be addressed to elevated mixing ratios of $RO_2$, which are reflected in the enhanced ROOH and the estimates of $RO_2 + HO_2$ by Tadic et al. (2020). Despite

matching results for the net photochemical production in some regions, the observations of $H_2O_2$ reflect less diurnal variation and overall lower mixing ratios than simulated by EMAC. This implies that further loss processes, e.g. the deposition during daytime and in general heterogeneous chemistry remain a major uncertainty in the photochemical budget of $H_2O_2$ in the MBL. We therefore emphasize the importance of $H_2O_2$ and organic peroxide in situ measurements, which were valuable to evaluate simulated deposition velocities and the accuracy of $HO_x$ simulations and lead to the assumption of a missing sink of $HO_x$ in

the model.

*Data availability*. All AQABA data sets used in this study are permanently stored in an archive on the KEEPER service of the Max Planck Digital Library (https://keeper.mpdl.mpg.de; last access: 28 April 2022) and are available to all scientists, who agree to the AQABA data protocol.


*Supplement*. The supplement to this article is available online at: DOI

*Author contributions*. DD, BB, HF and JL designed and supervised the study and the AQABA campaign. DD performed the HCHO and hydroperoxide measurements during the second leg of the campaign and evaluated the HCHO dataset. BB

performed the HCHO and hydroperoxide measurements during the first leg and provided the hydroperoxide dataset. HH, MM, ST and RR performed the LIF OH and $HO_2$ measurements during AQABA. JS and JNC performed the actinic flux measurements and calculated photolysis rates. PGE and JNC carried out the $O_3$ measurements during the cruise. AP performed the EMAC model runs.



*Competing interests*. The authors declare that they have no conflict of interest.

*Acknowledgements.* We thankfully acknowledge the cooperation with the Cyprus Institute (CyI), the King Abdullah University of Science and Technology (KAUST) and the Kuwait Institute for Scientific Research (KISR). We thank Hays Ships Ltd, Captain Pavel Kirzner and the *Kommandor Iona's* ship crew for the great support during all weather or wavy conditions and
for an unforgettable time on board. We would like to thank especially Marcel Dorf and Claus Koeppel for the organization of the campaign and Hartwig Harder for the management on board. Last but not least we are grateful for the whole AQABA community and a successful campaign.

*Financial Support.* The article processing charges for open-access publication were covered by the Max Planck Society.

*Review statement.*

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
