# Peer review of "Formaldehyde and hydroperoxide distribution around the Arabian Peninsula – evaluation of EMAC model results with ship-based measurements"

_Atmospheric Chemistry and Physics, 2022_

## Author Comment (AC1)

**Response to Referee 1:**

We would like to thank Referee 1 for the detailed review of our manuscript and the highly valuable feedback. Changes of the manuscript are highlighted in blue and the referee's comments in black.

In "Formaldehyde and hydroperoxide distribution around the Arabian Peninsula – evaluation of EMAC model results with ship-based measurements", the authors describe observations of these species from the AQABA campaign. They find that, for formaldehyde, agreement between observations and EMAC is acceptable, although there is a notable low bias in the model in polluted regions. Agreement for hydroperoxides is worse and is driven in large part to errors in OH and HO2 abundance. They also compare deposition velocities determined from observations to model output, finding reasonable agreement for both HCHO and H2O2 over the Arabian Sea but large underestimates by the model in the Mediterranean. In general, this is a well-written, thorough paper and is suitable for publication in ACP once the following minor comments are addressed.

Line 59: Need a space between "collision" and "partner"

Now corrected in the manuscript.

Line 115: "Distinguish" is misspelled

Now corrected in the manuscript.

Line 141: The statement "Tropospheric HCHO, which is not removed heterogeneously via deposition" is incorrect and contradicted in the next paragraph where you discuss heterogenous loss of HCHO (Lines 158 – 159). Please remove or clarify your point.

Now corrected in the manuscript:

Line 145: 'Photochemical losses of HCHO are the reaction with OH and its photolysis (R15 – R17) (e.g. Heikes et al., 2001).'

Lines 161 – 163: 'Wet and dry deposition are major loss processes of HCHO, even though it is less soluble than $H_2O_2$. Highest mixing ratios of HCHO are typically found in the boundary layer and decrease with altitude in the free troposphere (Zhu et al., 2020; Anderson et al., 2017; Stickler et al., 2007).'

Line 191: Later on, you mention that you correct for line losses of HCHO. Is there any concern, given the "sticky" nature of HCHO, that you will also get HCHO sticking to your inlet tubing and then later desorbing? This could artificially bias your results high, although it would likely only matter in clean background air with low concentrations. Does the relatively large flow rate prevent this?

Theoretically, these desorbing effects could bias the results in clean conditions with rapidly changing HCHO mixing ratios, e.g. when detecting ship emission plumes during rather clean MBL conditions of the Arabian Sea. Desorption should lead to elevated / variable zero air measurements, which was not observed.

Additionally, the determination of sampling artifacts was performed based on a HCHO permeation source, with gaseous HCHO standard injections at the start of the sampling line compared to injections

right in front of the AL4021 instrument. No significant losses were detected and the HCHO mixing ratio declined fast to stable ambient air levels. We did not observe any significant desorbing effects or longer decay intervals of HCHO mixing ratios after these HCHO standard injections.

Line 206: Does that mean Sect. 2.5? Check the ACP formatting guidelines, but I think you need say Section explicitly. As it's written, it's slightly confusing.

Now corrected in the manuscript.

Section 2.6: Given the importance of the OH and $HO_2$ measurements to your analysis, I think more discussion about these observations is warranted, particularly as they relate to the instrumental uncertainty/accuracy. Measurement uncertainty of these species can be relatively large and could have an impact on your comparisons.

Added more information about the OH and $HO_2$ measurements and the data quality:

Lines 270 – 275: 'The instrument utilizes LIF of the OH radical at 308 nm, which is created by a pulsed tunable laser system (Nd:YAG) operated at a pulse frequency of 3 kHz. OH radicals are excited in a low pressure detection cell (White cell setup) with a flow rate of 10 L min$^{-1}$. The detection of $HO_2$ is achieved by chemical conversion by the addition of NO downstream of the OH detection. The resulting sum of ambient plus chemically converted OH is measured in a second detection axis (Hens et al., 2014; Marno et al., 2020). HORUS achieved an instrumental LOD of 0.03 – 0.15 pptv for OH and 0.22 – 2.01 pptv for $HO_2$ with a TMU of 17 % (OH) and 20 % ($HO_2$), respectively.'

Line 289: How does this coarse grid resolution affect your results? The land/water interface is a difficult region to capture accurately, even at high resolution. Why was this resolution chosen (I assume you were using a model run for a different purpose than just for this paper)?

The AQABA dataset is ideal with its complex photochemistry in the MBL to examine the accuracy of EMAC's complex chemistry mechanism (MOM) and the involved submodels (Pozzer et al., 2022; https://doi.org/10.5194/gmd-15-2673-2022). Please keep in mind that EMAC simulates the atmosphere globally, and thus its resolution is still limited. Even though I agree that a higher resolution would be a benefit for the study, e.g. if the model would be able to resolve the air pollution over the Arabian more accurately with a higher resolution. A short comparison was also done with a higher resolution based on the regional model WRF-Chem, but the increased resolution did not improve the results (e.g. over the Arabian Gulf) and was thus not included in the manuscript.

Line 293: What emissions inventory are you using? Have there been evaluations of these inventories before, particularly in the middle east?

The EMAC results were based on EDGAR (v4.3.2) see also Pozzer et al. (2022). The AQABA campaign demonstrates the first ship-based measurements over the Arabian Gulf, as far as I am aware of. To the best of my knowledge I do not know of any evaluations of emission inventories within the region.

Line 299 – 301: 'The anthropogenic emissions are based on the Emissions Database for Global Atmospheric Research (EDGARv4.3.2). Further details are presented in Pozzer et al. (2022).'

Line 314: The technique you are using to determine the deposition velocity is highly dependent on your assumption that there is very little spatial heterogeneity in the concentrations of either HCHO or H2O2. I think some further analysis is needed to show that this is the case, either here or in Section 3.4. Do you, for example, remove any plumes that you might encounter at nighttime? What distance did the ship cover in a night? Is it a large enough distance that differences in surface conditions (winds, waves, etc.) could affect the deposition velocity? How can you be sure that you're not sampling air that has recently been advected from a region with a different background HCHO value?

Plumes of HCHO (e.g. due to ship emissions) have been removed with the use of $NO_x$, CO, $SO_2$ and wind direction data (Lines 344 – 346 and caption of Fig. 2). No significant plumes were detected for $H_2O_2$. This analysis was only performed over the rather clean Arabian Sea and over the Mediterranean Sea, where we detected well aged air masses.

HYSPLIT trajectories (Section 2.8) showed that we measured air masses with a similar origin over the Arabian Sea, while winds over the Mediterranean Sea originated in Europe (Fig. S17). The *Kommandor Iona* moved with an average speed of $3.4 \pm 1.8$ m s$^{-1}$, while we were sailing slowest during the high wave conditions over the Arabian Sea and the Mediterranean Sea ($\leq 10$ km h$^{-1}$). We cannot exclude the possibility that high wave conditions affected our results e.g. due to high amounts of sea spray.

Figure 1: Was there any particular rationale as to how you divided the regions up, beyond names, particularly for the RN and RS? Were the chemical environments significantly different or was this just an arbitrary decision?

The regions were divided according to changes in chemical conditions and changes of the air mass origin based on HYSPLIT trajectories. We added 'according to different chemical regimes' to the caption of Fig. 2. Additionally, air mass origin plots based on HYSPLIT trajectories were added to the supplement (Fig. S17, S18), HYSPLIT is described in section 2.8.

Figure 2 caption: Should be "Contaminated HCHO data were removed"

Now corrected in the manuscript. Additionally, the readability of Fig. 2 was improved by increasing the size of the captions and its resolution.

Line 389: If MS and RS were relatively clean regions, why do you think you saw EHP there?

Thanks for this input, we agree that air masses over the MS were not clean compared to background MBL conditions. We changed the description of the chemical regime over the MS, which reflected mainly well-aged air pollution transported from Europe.

Line 407 – 409: Significantly enhanced amounts of EHP were only detected over the Arabian Gulf, although small amounts of EHP were also detected in the enriched samples of MS (Fig. S10), where we detected aged air masses originating from Europe (Fig. S17).

Line 405: How were the EMAC data adapted? Linear/bilinear interpolation? Did you interpolate in space and time? What was the time resolution of the model output?

The model results were interpolated bilinearly along the ship track (GPS data) with the S4D submodel (Jöckel et al., 2010). No interpolation is present on the time axis.

Line 294 – 300: 'The model simulations were carried out in the T106L31 resolution, which correspond to a grid of 1.1° · 1.1° (~110 km) with 31 vertical pressure layers and a time resolution of 10 minutes. The EMAC data was interpolated bi-linearly along the GPS track of the ship with the S4D submodel (Jöckel et al., 2010). The model was initialized from a previous evaluated simulation (Pozzer et al., 2022) and started on the 1st of June 2017 covering the entire campaign. The dynamics have been weakly nudged (Jeuken et al., 1996; Jöckel et al., 2006) towards the ERA-interim data (Berrisford et al., 2011) of the European Centre for Medium-Range Weather Forecasts (ECMWF) to reproduce the actual day-to-day meteorology in the troposphere.'

Line 412 (Fig. S2): It would be helpful to color code figure S2 by region to emphasize your point.

Thanks for the idea, we included updated versions of the scatter plots in the SI.

Figure 3: Could the persistent low bias in HCHO in the polluted regions from EMAC also result from the coarse model resolution?

It seems unlikely that the elevated HCHO mixing ratios over the AG and SU were only due to the coarse model resolution and the resulting continental influence in some grid cells. There is a distinct trend for higher amounts of air pollution during the first leg over the AG and SU for both the HCHO measurements and the EMAC results, while the second leg was less polluted in both datasets (Fig. S1). Even though we agree that a higher resolution of EMAC would be helpful to resolve more details.

Line 428: Should be "Even though".

Now corrected in the manuscript.

Line 436: How well does the model capture NOx in the more polluted regions? Could errors in NOx abundance also affect the model HCHO accuracy there? If modeled NO is too high in polluted regions, would production from the CH3O2 + NO reaction also be too high, in which case you would be producing HCHO from the wrong source?

According to Tadic et al. (2020) median $NO_x$ values over the Arabian Gulf matched within a factor of less than 2 with 1.26 ppbv for the measurements and 1.61 ppbv for the EMAC simulation. The averages show that we observed a higher variation of $NO_x$ than the model simulated, with 3.65 ppbv measured and 1.91 ppbv simulated. EMAC was of course not able to resolve fine details, as is the case for $O_3$ and HCHO due to its coarse resolution, still it simulated satisfactory results for $NO_x$ and matched the observations within a factor of 2.

Line 437: If you just look at daytime values, does the measurement/model agreement change at all, since the model does have a pronounced diurnal cycle?

In a previous version of the manuscript these plots were separated into photochemistry (yellow) and nighttime (grey). As there is no general trend that the accuracy is higher during night or day, we decided to remove these plots. The enhanced background concentrations of HCHO over the Arabian Sea were likely caused by ship emissions since we were sailing on a major shipping route (e.g. Line 378).

[Figure]

Line 461: I might consider breaking Section 3.2 into parts, one for HCHO and one for the other species. It's very long otherwise. Also, I feel like you contradict yourself in the sentence where you say "The model-measurement comparison for H2O2 is even worse…". You say in the previous sentence that HCHO "agreement is quite good." I would reword one of those sentences. (I feel like "quite good" might be an overstatement given the regressions you show in the supplement, but, given the limitations, the model is at least satisfactory).

Thanks for the idea, but we would like to keep the major structure as it simplifies comparisons between the HCHO and $H_2O_2$ observations within the discussion.

Line 476 – 477: 'Given the multitude of potential HCHO sources both from direct emissions and a large variety of photochemical precursors, and the limited resolution of EMAC the agreement within a factor of 2 is satisfactory.'

Line 485 – 486: 'The model-measurement comparison for $H_2O_2$ reveals that EMAC systematically overestimates $H_2O_2$ mixing ratios by up to an order of magnitude.'

Line 465: Couldn't this also be due to an underestimate in the magnitude of the sinks?

Line 488 – 489: 'This consistent overestimation by the model indicates either a significant overestimation of $H_2O_2$ sources or missing sinks in the model, or a combination of both.'

Figure 6: How accurate are the meteorological variables in EMAC? Do you find general agreement between observed temperature and pressure (the variables that affect reaction rate) and the model? If there are significant land/water differences, given the grid size, this could impact the accuracy of the loss rate calculations.

We agree, and additionally checked for the accuracy of water, which is influences the production of $H_2O_2$ through recombination of $HO_2$. EMAC shows a slight offset of the temperature when cruising close to the coastline of Egypt (RN) and Oman (GO) probably due to the land / water difference. These only marginally affected the reaction rates, but the underestimation of $H_2O$ led to an underestimation of

$k_{HO2+HO2}$. Main cause of the overestimated $H_2O_2$ remains the overestimated $HO_2$, underestimated dry deposition during the day may be possible, but remains unclarified.

---

## Author Comment (AC2)

**Response to Referee 2:**

We would like to thank Referee 2 for the detailed review of our manuscript and the valuable feedback. Changes of the manuscript are highlighted in blue and the referee's comments in black.

Referee 2:

The manuscript by Dienhart et al. uses ship-based measurements of HCHO, $H_2O_2$, ROOH, OH, and $HO_2$ as well as actinic flux measurements in order to assess the oxidative budget of the marine boundary layer (MBL) around the Arabian Peninsula in Summer 2017. With such limited measurements over the open ocean in the literature, the authors possess a valuable dataset to assess model performance over such regions. The authors use their measurements to assess the general circulation model EMAC and also report nighttime deposition velocity measurements of HCHO and $H_2O_2$ over the Mediterranean and Arabian Seas.

General comments: Ship-based measurements are difficult to perform, and so it is very impressive the authors were able to collect a dataset during AQABA of HCHO, $H_2O_2$, organic hydroperoxides (ROOH), OH, and $HO_2$. A really decent job was done characterizing the uncertainty of their measurements given the circumstances of being on a ship. More will be discussed in the major revisions section below, but the lack of performing sensitivity tests with the EMAC model (such as by scaling VOC or other ship emissions to see impact on $HO_x$ and subsequently $H_2O_2$ and HCHO) made the model-measurement comparison of the paper weak. Additionally, several substantial questions arose while reading Section 3.3 when the authors use the net photochemical production/loss of $H_2O_2$ to evaluate the discrepancy between measured and modeled $HO_x$. At times, the manuscript reads as a "Measurement Report" as opposed to a "Research Article" for ACP.

While the manuscript fits within the scope of ACP and presents a unique dataset for assessing oxidation chemistry in the MBL, major revisions are necessary when performing the model-measurement comparison, and so I would only recommend publication after the below issues are fully addressed. Authors could also consider turning the manuscript into a Measurement Report.

Major Revisions:

Abstract: The abstract does not report any quantitative conclusions from the study with several sentences better suited for the introduction as opposed to an abstract. For example, the authors could report their determination of the nighttime deposition velocities of H2O2 and HCHO over the Mediterranean and Arabian Seas since that would summarize their findings from Section 3.4.

We agree and changed the abstract:

'Formaldehyde (HCHO), hydrogen peroxide ($H_2O_2$) and organic hydroperoxides (ROOH) play a key role in atmospheric oxidation processes. They act as sources and sinks for $HO_x$ radicals (OH + $HO_2$), with OH as the primary oxidant that governs the atmospheric self-cleaning capacity. Measurements of these species allow evaluation of chemistry-transport models which need to account for multifarious source distributions, transport, complex photochemical reaction pathways and deposition processes of

these species. HCHO is an intermediate during the oxidation of VOCs and is an indicator of photochemical activity and combustion related emissions. In this study, we use in situ observations of HCHO, $H_2O_2$ and ROOH in the marine boundary layer (MBL) to evaluate results of the general circulation model EMAC (ECHAM5/MESSy2 Atmospheric Chemistry). The dataset was obtained during the AQABA ship campaign around the Arabian Peninsula in summer 2017. This region is characterized by high levels of photochemical air pollution, humidity and solar irradiation, especially in the areas around the Suez Canal and the Arabian Gulf. High levels of air pollution with up to 12 ppbv HCHO, 2.3 ppbv ROOH and relatively low levels of $H_2O_2$ ($\leq$ 0.5 ppbv) were detected over the Arabian Gulf. We find that EMAC failed to predict absolute mixing ratios of HCHO and ROOH during high pollution events over the Arabian Gulf, while it reproduced HCHO on average within a factor of 2. Dry deposition velocities were determined for HCHO and $H_2O_2$ during night with $0.77 \pm 0.29$ cm s$^{-1}$ for HCHO and $1.03 \pm 0.52$ cm s$^{-1}$ for $H_2O_2$ over the Arabian Sea, which were matched by EMAC. The photochemical budget of $H_2O_2$ revealed elevated $HO_x$ radical concentrations in EMAC, which resulted in an overestimation of $H_2O_2$ by more than a factor of 5 for the AQABA dataset. The underestimated air pollution over the Arabian Gulf was related to EMACs coarse spatial resolution and missing anthropogenic emissions in the model.'

Line 141: Authors contradict themselves when they state that HCHO "is not removed heterogeneously via deposition" and then in Lines 158-159 say "heterogeneous losses via wet and dry deposition also significantly influence the HCHO distribution". Line 141 is simply incorrect and should be fixed.

Now corrected in the manuscript:

Line 145: 'Photochemical losses of HCHO are the reaction with OH and its photolysis (R15 – R17) (e.g. Heikes et al., 2001).'

Lines 161 – 163: 'Wet and dry deposition are major loss processes of HCHO, even though it is less soluble than $H_2O_2$. Highest mixing ratios of HCHO are typically found in the boundary layer and decrease with altitude in the free troposphere (Zhu et al., 2020; Anderson et al., 2017; Stickler et al., 2007).'

Section 2.7: Authors must mention how EMAC was initialized AND what was the spin uptime used in their model runs. Additionally, since HCHO is the tracer generally used to assess recent VOC oxidation, I was surprised the authors had no plots of whether EMAC over or underestimated the VOCs measured during AQABA (Line 290 suggests VOC measurements exist from the campaign). Seeing the VOC data would be helpful when looking at the model-measurement discrepancies of HCHO.

We extended the description of the model run (Lines 294 – 303):

'The model simulations were carried out in the T106L31 resolution, which correspond to a grid of $1.1° \cdot 1.1°$ (~110 km) with 31 vertical pressure layers and a time resolution of 10 minutes. The EMAC data was interpolated bi-linearly along the GPS track of the ship with the S4D submodel (Jöckel et al., 2010). The model was initialized from a previous evaluated simulation (Pozzer et al., 2022) and started on the 1$^{st}$ of June 2017 covering the entire campaign. The dynamics have been weakly nudged (Jeuken

et al., 1996; Jöckel et al., 2006) towards the ERA-interim data (Berrisford et al., 2011) of the European Centre for Medium-Range Weather Forecasts (ECMWF) to reproduce the actual day-to-day meteorology in the troposphere. The anthropogenic emissions are based on the Emissions Database for Global Atmospheric Research (EDGARv4.3.2). Further details are presented in Pozzer et al. (2022). Previous results of airborne and shipborne expeditions have been compared to EMAC (Fischer et al., 2015; Klippel et al., 2011), also the AQABA datasets of $NO_x$, $O_3$ and VOCs during AQABA have been published (Tadic et al., 2020; Wang et al., 2020).'

Thanks for the idea of implementing the VOC measurements in the study, but these have already been published and compared to EMAC (see Wang et al., 2020; https://doi.org/10.5194/acp-20-10807-2020)

Lines 330-332: The logic is inconsistent. If it was deemed that HCHO data was contaminated by the ship exhaust plumes and thus should be filtered out, then the same filtering should apply to all other measurements. The authors mention that this contaminated air mass had high $NO_x$ levels, which would definitely have impacted the oxidative regime the instruments were sampling at that time. It would be an unfair comparison with EMAC to use measurement data that was known to be contaminated by Kommandor Iona because the model would not be expected to know those ship emissions.

Thank you for this input, we agree that this comparison with EMAC is unfair, but still we decided to keep the analysis like it is. As mentioned in the manuscript there is no evidence that the stack emissions affected the peroxide measurements despite the known (marginal) $NO_x$ interference of the instrument (2 pptv $H_2O_2$ on average with a maximum interference of 48 pptv $H_2O_2$). Compared to the highly elevated mixing ratios in EMAC the effect on the comparison is negligible, it would not change the outcome to perform the analysis again. Other than that, the ship emission filter was applied for the calculation of the photochemical budget of $H_2O_2$.

Line 455: It is vague and qualitative to just conclude that "VOC emissions in the Arabian Gulf need to be adjusted". How much do the emissions need to change? Is the necessary emission change even reasonable? Since the crux of the paper is the model-measurement comparison as stated in the title, then a sensitivity test should be performed and its results fully described by raising VOC emissions in EMAC by say 10% (or some number) and seeing whether that helps to close the discrepancy between the EMAC model and HCHO measurements and how the increased VOC emission impacts model oxidants like OH and $HO_2$.

In fact, no sensitivity tests were performed with the model on any regions in the manuscript. Evaluation of model performance as implied by the title would imply that the model would be run with different initial conditions or settings to see the impact on oxidants.

We agree and changed lines 473 – 479:

'The model simulates the AS cleaner than it was observed, but the relatively low levels of HCHO over RS with winds from Eritrea (Fig. S2, S17) were overestimated. Analysis of the air mass origin showed that air masses over the Arabian Sea represented clean and aged conditions transported from the center

of the Indian Ocean (Fig. S18). Given the multitude of potential HCHO sources both from direct emissions and a large variety of photochemical precursors, and the limited resolution of EMAC the agreement within a factor of 2 is satisfactory. The comparison of simulated HCHO based on a higher resolved model (WRF-Chem) did not improve the accuracy of HCHO and was thus not included in the manuscript.'

Thank you for the idea of the addition of a sensitivity test. We agree that it would improve the results, but in our opinion it exceeds the scope of this draft. With a regional or box model a sensitivity test would be highly valuable and less complex to perform. The scope of the manuscript was to analyze the oxidation budget in EMAC and compared it to the observations, not the perform a model optimization. The AQABA measurements were additionally compared to a regional model with an improved spatial resolution (WRF-Chem, ~10km grids). The higher spatial resolution did not improve the results over the Arabian Gulf for HCHO, but were based on a different emission database. Thus we decided not to include them in this manuscript. Some regions are improved compared to EMAC, probably due to the higher resolution, but the polluted regions were not matched:

[Figure]

Section 3.3: The authors are attempting to use $H_2O_2$ to evaluate the discrepancy between measured and modeled $HO_x$ since the production of $H_2O_2$ depends quadratically on $HO_2$ and its photochemical loss linearly on OH. They want to do this by only looking at the effect of photochemistry on the $H_2O_2$ mixing ratio (Line 547), or said another way, completely ignoring $H_2O_2$ deposition (ignoring the last term in Equation 4).

While there's nothing intrinsically incorrect with Figure 6, I believe there's a grave error in Figure 7 when the authors have to multiply the loss rate constants by the $H_2O_2$ mixing ratio (Line 546). The moment a person multiplies by the $H_2O_2$ mixing ratio, the assumption that the authors are only looking

at effect of photochemistry on $H_2O_2$ production and loss breaks down since the $H_2O_2$ mixing ratio is the net result of photochemistry, deposition, and transport. I'm not entirely sure if it's justified a priori to say $H_2O_2$ deposition is negligible (particularly when one looks at Figure 11; bottom panel for $H_2O_2$ nighttime deposition velocities).

Thanks for this input, we decided on purpose to display the losses of $H_2O_2$ in Fig. 6 without the multiplication with the $H_2O_2$ mixing ratio, so that the loss rate constants could be compared. By multiplication with the mixing ratio this plot only shows the obviously strong offset by EMAC. In Fig. 7 these lose rate constants were multiplied by the $H_2O_2$ mixing ratios before the subtraction.

The manuscript does not state that we want to neglect dry deposition of $H_2O_2$ during the day! In fact, we think that it might be a major issue for the discrepancy between EMAC and the observations besides the overestimation of $HO_x$. The determination of the dry deposition during the day was not possible based on our observations, only an estimation could be done based on the PSS assumption and thus we cannot compare it to the model. Even though, it seems likely that dry deposition during the day may be more effective than during night e.g. due to enhanced turbulence.

To make this clearer, we changed some sentences and added more information to the captions of Fig 6. Lines 562 – 563:

'Besides dry deposition, photochemical losses of $H_2O_2$ are the reaction with OH and photolysis, which were calculated according to Eq. 4, and displayed without multiplication of the $H_2O_2$ mixing ratio for the sake of comparability (Fig. 6).'

Lines 569 – 570:

'Loss due to dry deposition could not be determined during the day, but nighttime deposition velocities are calculated in section 3.4.'

Lines 572 – 574:

'Please note that in this term neither physical loss processes (e.g. deposition) nor transport are represented and thus it only reflects the effect of local photochemistry on the $H_2O_2$ mixing ratio.'

There's also an interesting case with the Mediterranean Sea (MS) in Figures 7 and 8 where the net photochemical production agrees yet there are significant model-measurement discrepancies between $H_2O_2$, OH, and $HO_2$ mixing ratios in Figures 4 and 5. The authors only say that overestimated model photochemical production of $H_2O_2$ in MS is compensated by elevated losses via reaction with OH (Lines 548-549) so that both model and measurement agree, but could the authors elaborate more and specifically say what's wrong with the $HO_x$ budget in the model based solely on looking at MS in Figure 7? Related questions could be asked as well about the other regions.

We agree that there are significant overestimated levels of both OH and $HO_2$ (Fig. 4 and 5), yet the overestimation of $HO_2$ over the MS is not as dramatic compared to other regions. The surprisingly well matched net photochemical production of $H_2O_2$ over the MS is caused by overestimated OH which compensates the too high levels of $HO_2$. A similar case are the results for the net photochemical

production over the northern Red Sea. We agree that the question arises: 'Why is there a drastic offset in $H_2O_2$ while net photochemical production between observations and EMAC match?'.

Many physical and heterogeneous processes remain uncertain, e.g. we cannot exclude that we experienced additional loss due to sea spray during the high wave conditions over the MS. Neither can we exclude the effect of transport. We checked for vertical entrainment in EMAC, which would lead to drastic enhanced levels of $H_2O_2$ in the early morning when the BLH increases. This may have affected the results, but it does not explain the dramatic overestimation of $H_2O_2$ over the MS where EMAC simulates a rather stable MBL. Furthermore, we mostly measured in chemical regimes which contained elevated $NO_x$ which likely suppressed peroxide formation. Additionally, we compared OH production in the model to determine on which reactions the overestimated $HO_x$ is based on. It matches best over the MS, while the main cause of the overestimated $HO_x$ is an overestimation of $O_3$ and / or missing losses. Moreover, we observed decreased $HO_2$ and OH over the Arabian Gulf, which was likely related to enhanced $RO_2$ radicals which slowed down OH recycling (not matched by EMAC).

[Figure]

Revisions:

-        Lines 344-352: The reader would be greatly aided by a wind vector plot that shows the direction of where the air mass was coming from at each point for each leg of the campaign. For instance, when looking at points in the Arabian Sea (say in Figure 2), the reader would want to know whether the air mass that was being sampled originated from the open ocean or whether it was continental outflow that maybe was contaminated by other VOCs.

We included air mass origin plots based on HYSPLIT trajectories for the eight separated regions in the SI of the manuscript (Fig. S17, S18, section 2.8). Additionally, the readability of Fig. 2 was improved by increasing the size of the captions and its resolution.

- Lines 354-358: I think more has to be said than just comparing the magnitude of the HCHO mixing ratio to other marine locations. Are the VOCs around the Arabian Peninsula generally larger than the central Indian Ocean (Wagner et al 2001) or the tropical Atlantic (Weller et al 2000)? HCHO is highly dependent on localized VOCs.

Added additional information in line 372 – 379:

'Wagner et al. (2001) performed ship-borne measurements during the INDOEX campaign in the central Indian Ocean with HCHO mixing ratios between 0.2 – 0.5 ppbv, with the lowest mixing ratios in the clean maritime background of the southern hemisphere and about 0.5 ppbv HCHO in continentally influenced air masses. Weller et al. (2000) reported ship-based HCHO measurements in the Atlantic, which reached a broad maximum with values of 1.0 – 1.2 ppbv in the tropical Atlantic, decreasing towards the poles with values below 0.8 ppbv. These air masses represented pristine MBL conditions with average daytime NO of 3.1 pptv. During AQABA we did not encounter such very clean conditions with the lowest median $NO_x$ of 0.19 ppbv for AS and 0.25 ppbv for MS. This was likely due to sailing on major ship traffic routes which may have also led to enhanced background HCHO compared to remote MBL conditions.'

- Line 410: "most likely due to missing localized pollution events": It would be helpful to plot out the model-measurement differences along the ship tracks (similar presentation as Figure 2) since this would show whether there was a missing localized pollution event. Conversely, pointing out some examples in the SI plots would be helpful, but the spatial information is lost.

Changed the discussion and added a reference to Paris et al. (2021) who showed that gas flaring emissions were a major source of the elevated VOCs. Moreover, the pollution event over the AG is now presented in the SI with a GPS plot and HYSPLIT trajectories.

Lines 439 - 454:

'Elevated $NO_x$ and $O_3$ measurements classified AG and SU as the most polluted regions, followed by RN and GO which both were influenced to a higher extent by anthropogenic pollution (Tadic et al., 2020). The elevated HCHO during the high ozone and VOC conditions on the first leg in AG was not reproduced by EMAC. Here the model clearly underestimates HCHO (Fig. S1) and ROOH (Fig. S5). Since the elevated OVOCs correlated well with CO and $O_3$ (Wang et al. 2020), we can assume that we probed a highly polluted and photochemically active air mass, with both effective photochemical production and primary emissions of HCHO. Paris et al. (2021) identified natural gas flaring as a major source of the elevated VOCs over the Arabian Gulf. During this event, maximum HCHO and ROOH mixing ratios were measured during AQABA, with values up to 12.6 ppbv HCHO and 2.26 ppbv ROOH in the center of the Gulf (Fig. 2, S9, S19). The event was less pronounced in EMAC with up to 3.31 ppbv HCHO leading to an underestimation over the AG by about a factor of 4. EMAC does not simulate significantly elevated values of ROOH peaking at 0.49 ppbv with elevated contributions of PAA and EHP. Even though, the model underestimates ROOH also by about a factor of 4 (Fig. S9). Wang et al.

(2020) showed that EMAC simulates enhanced acetone and methyl ethyl ketone (MEK) during this event, although the model shows no significant increase of acetaldehyde.

In the Suez Canal and the Gulf of Suez (SU), the second most polluted region, the model also underestimates HCHO mixing ratios, even though EMAC simulates a significant increase of HCHO compared to MS and RN. Wang et al. (2020) showed that these air masses have been influenced by biomass burning and increased anthropogenic emissions e.g. by gas flaring similar to the Arabian Gulf.'

- Line 455 and Line 461: Avoid qualitative statements like "the agreement is quite good"or "even worse". It's completely subjective.

Changed lines 471 – 479:

'Altogether, EMAC reproduces observed HCHO mixing ratios on average within a factor of 2. However, the model clearly underestimates air pollution over the Arabian Gulf, which leads to the assumption of missing sources in EMAC and may also be related to the limited resolution of EMAC. The model simulates the AS cleaner than it was observed, but the relatively low levels of HCHO over RS with winds from Eritrea (Fig. S2, S17) were overestimated. Analysis of the air mass origin showed that air masses over the Arabian Sea represented clean and aged conditions transported from the center of the Indian Ocean (Fig. S18). Given the multitude of potential HCHO sources both from direct emissions and a large variety of photochemical precursors, and the limited resolution of EMAC the agreement within a factor of 2 is satisfactory. The comparison of simulated HCHO based on a higher resolved model (WRF-Chem) did not improve the accuracy of HCHO and was thus not included in the manuscript.'

- Line 441: Instead of assuming, can it be shown that the air mass encountered in the Arabian Sea during AQABA originated from the continent, some industrial area, a ship plume, etc.? Can you run a Lagrangian model to get a back trajectory of the air mass? This is important since HCHO over the remote ocean is generally from $CH_4$ oxidation.

Added HYSPLIT trajectories for the respective regions to the SI (Fig. S17, S18, section 2.8).

- Line 582: Remove "transport" from the section title as it isn't discussed.

Now changed in the manuscript.

Technical Corrections:

Thanks for the identification of these typos, we changed them in the new version of the manuscript.

- Line 150: Add "recent emissions from" right before the word "anthropogenic activity".

- Line 414: Change "Read" to "Red"

- Line 429: I think the authors meant "though" instead of "tough"?

- Line 507: 16.18.2017 is not a valid date

- Figure 6, 7, 8: Use of $L(H_2O_2)$ notation. The use of this notation in these figures is not the same as defined in Equation 4 since it doesn't include the deposition term. Either change the notation in Equation 4 or somehow denote that this is photochemical only on the figures themselves.

Thanks for this input, we changed the notation in the figures and added more information in the discussion that the deposition is still missing for the calculated net photochemical production.

- Figure 10: Legend needed on all subplots

Now changed in the manuscript.